# Early Oldowan technology thrived during Pliocene environmental change in the Turkana Basin, Kenya

Approximately 2.75 million years ago, the Turkana Basin in Kenya experienced environmental changes, including increased aridity and environmental variability. Namorotukunan is a newly discovered archaeological site which provides a window into hominin behavioral adaptations. This site lies within the upper Tulu Bor and lower Burgi members of the Koobi Fora Formation (Marsabit District, Kenya), presently a poorly understood time interval due to large-scale erosional events. Moreover, this locale represents the earliest known evidence of Oldowan technology within the Koobi Fora Formation. Oldowan sites, older than 2.6 million years ago, are rare, and these typically represent insights from narrow windows of time. In contrast, Namorotukunan provides evidence of tool-making behaviors spanning hundreds of thousands of years, offering a unique temporal perspective on technological stability. The site comprises three distinct archaeological horizons spanning approximately 300,000 years (2.75 – 2.44 Ma). Our findings suggest continuity in tool-making practices over time, with evidence of systematic selection of rock types. Geological descriptions and chronological data, provide robust age control and contextualize the archaeological finds. We employ multiple paleoenvironmental proxies, to reconstruct past ecological conditions. Our study highlights the interplay between environmental shifts and technological innovations, shedding light on pivotal factors in the trajectory of human evolution.

The remarkable ability of humans to inhabit nearly every terrestrial ecosystem is a result of the synergy between biological and technological evolution[1]. The long term significance of our evolutionary relationship with technology arises from the discovery of stone artifacts within Plio-Pleistocene sediments[2,3]. This study provides new evidence from the Turkana Basin for the relationship between climatic and environmental shifts and the development of stone tool technologies by hominins. The earliest known assemblages of Oldowan artifacts (~2.9-2.6 Ma) are confined to four localities in eastern Africa[4,5] (Fig. 1). The earliest known localities provide insights into technological behaviors at single time horizons in the deep past. Despite advances in our understanding of early human technology, the specific

mechanisms through which environmental changes influenced technological evolution in the earliest Oldowan remain poorly understood. Demonstrating how early hominins adapted their tool-making practices in response to changing environments would provide new insights into the evolutionary pressures that shaped these innovations. Here, we describe multiple assemblages of stone artifacts from well-constrained horizon, with age estimates at 2.75, 2.58, and 2.44 Ma, from the Koobi Fora Formation in the northeastern portion of the Turkana Basin (Paleontological Collection Area 40, archeological site Namorotukunan, National Museums of Kenya ID: FwJj 52). Detailed knowledge of environmental patterns within this region allows us to explore the interplay between periods of environmental change and

e-mail: David_braun@gwu.edu; dan.palcu@gmail.com

**Fig. 1 | Map of Turkana Basin with the Namorotukunan Archeological Site and timeline of currently known events in the Plio-Pleistocene. a** Geographical context of the Koobi Fora Formation (red stripes), the paleontological collection area 40 (green square), and the location of the site of Namorotukunan (black dot); [map produced Natural Earth and NOAAA ETOPO 2022[95]]; **b** Stratigraphic context of the Koobi Fora Formation highlighting members and key volcanic ash marker levels, yellow bars refer to the age of archeological horizons (tephrostratigraphy after McDougall et al.[96]); **c** A chronology of key Plio-Pleistocene hominins from the

East African Rift System (EARS)[11,74,97,98] **d** A chronology and key localities associated with hominin lithic technology[3,6,12] (images of Nyayanga provided by E. Finestone; images of Lomekwi and BD1 based on 3D models; artifact images are for representation and not to scale) and the investigations at Namorotukunan: red arrows represent the artifact levels in the archeological excavations (photos DRB), and colored circles (lettered A-G) represent geologic sections investigated to develop a synthetic stratigraphic column (presented in Figs. 2 and 3).

the presence of an early Oldowan techno-complex that first evolved in the late Pliocene, and emphasizes temporal continuity throughout the Early Pleistocene, in eastern Africa.

The earliest phases of tool manufacture, dating back to over 3.0 million years ago[6] (although see Archer[7]), highlight percussive technology, which is ubiquitous in hominin records and shared with other primates[8,9]. Tool use, associated with extractive foraging, is a recurring trait in some extant primates[10]. The oldest systematic production of sharp-edged stone artifacts, known as the Oldowan, is found in the hominin behavioral record at eastern African sites: Ledi-Geraru and Gona in the Afar Basin (2.6 Ma), Ethiopia, and Nyayanga in western Kenya (2.6-2.9 Ma)[3,11-13].

In this study, we describe the earliest known Oldowan technologies and their paleoenvironmental context in the Koobi Fora Formation (Fig. 1). The presence of butchery marks on bones within the Namorotukunan assemblage underscores the role of sharp-edged tools in the foraging behavior of these hominins and suggests that the development of Oldowan technology was associated with the exploitation of resources mediated by tool use[14]. Open habitats such as savannas and grasslands expand at the end of the Pliocene and may have facilitated an adaptive shift in hominins towards a regular exploitation of foods requiring the use of tools (e.g., USOs, bone marrow, meat)[15,16]. Here we examine three distinct temporal horizons containing evidence of sharp-edged tool technology spanning ~300,000 years. The consistent technological approaches across this interval suggest an enduring technological adaptation in the hominin lineage throughout the late Pliocene and earliest Pleistocene.

## Results
### Geological context
The Koobi Fora Formation (KF Fm.) refers to the sedimentary strata of the Omo Group on the eastern side of Lake Turkana and lies unconformably on Miocene and Pliocene volcanic rocks and associated sediments[17,18]. The Omo Group encompasses these and other Plio-Pleistocene lacustrine and fluvial deposits that fill a series of alternating half-grabens of the Omo-Turkana Basin, which in turn is part of the larger East African Rift System (EARS)[19,]. Excellent age control throughout the Omo Group (4.02 to -0.75 Ma) is provided by $^{40}Ar/^{39}Ar$ dates on volcanic ash layers, K/Ar ages on intercalated basalts and paleomagnetic stratigraphy[20]. The KF Fm. is divided into eight members (see Fig. 1), each delineated by a dated volcaniclastic tephra marker bed at the base of the member (except the basal Lonyumon Mbr[19].). In Area 40, where Namorotukunan is located, sedimentary strata are estimated to range roughly between 4.3 to 1.6 Ma[21,22], with a gap between 3.0 −2.5 Ma represented by the Burgi Unconformity[21,23,24]. Above this unconformity, a series of lake clays indicate a local change in tectono-sedimentary patterns and the Paleo-Lake Lorenyang transgression in the Turkana basin[21,24]. To the west of Namorotukunan, across the North Gele Fault[19], a younger sedimentary succession is exposed that includes at least the Upper Burgi and KBS Members, as evidenced by the presence of the KBS Tuff within that sequence[21].

Geological investigations focused on a 46-meter-thick stratigraphic interval examined in seven sections (Fig. 2). These sections are assigned to local geological units used for mapping around Namorotukunan (Supplementary Fig. 1). The lowermost unit in this framework is a tephra level (TB), identified as the Tulu Bor Tuff[21]

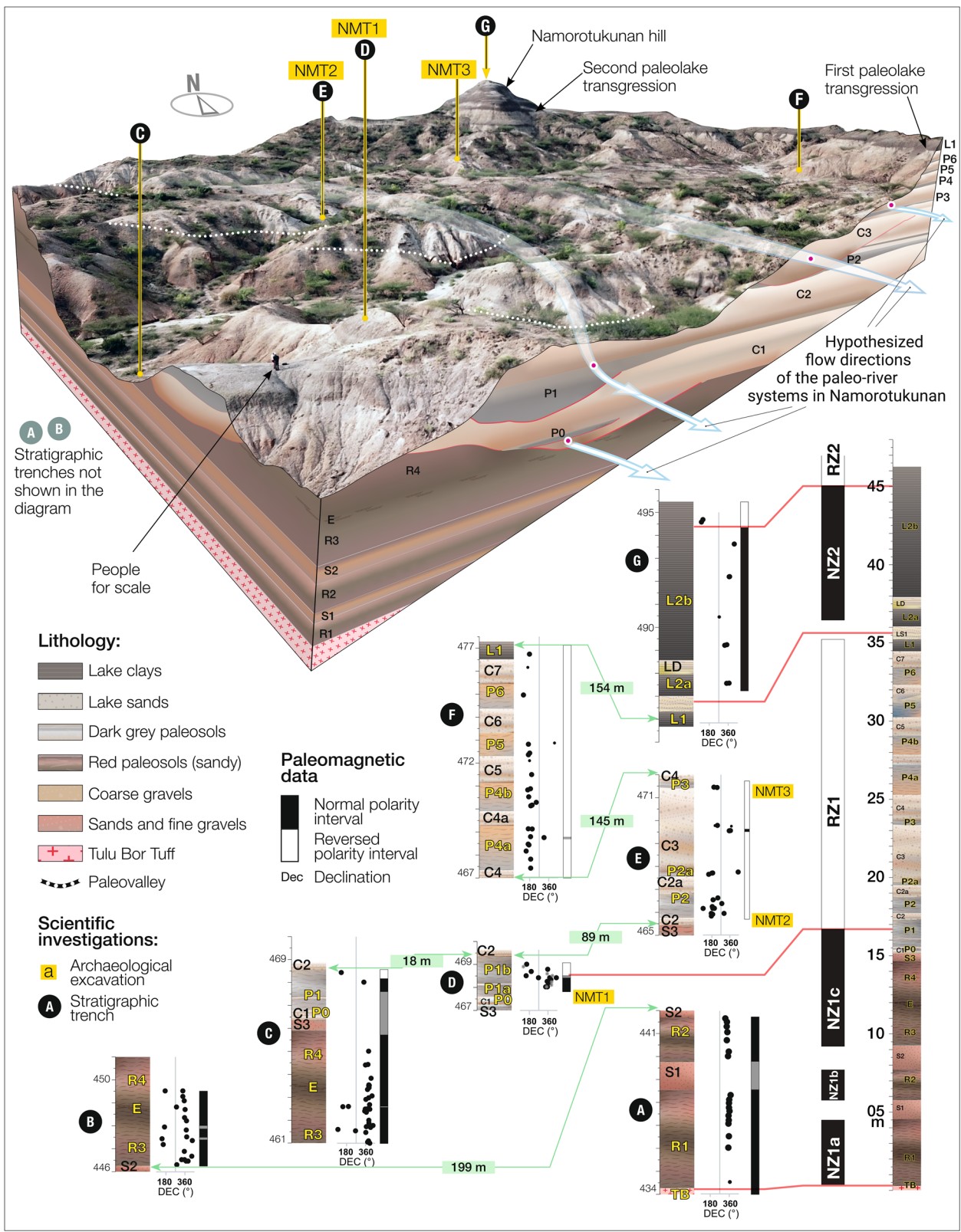

(Supplementary Table 1). The uppermost level is a sequence of lake clays associated with the hill locally known as Namorotukunan, from which the archeological site name is derived (L2b). This lithological succession comprises three distinctly different sediment packages separated by discrete boundaries (Supplementary Table 2).

The oldest sediment package is characterized by red silts, paleosols, and sands overlying the Tulu Bor Tuff (TB). It consists of poorly developed paleosols with root traces and carbonate-rich silts intercalated with sands and fine gravels. This package is interpreted as a fluvial floodplain (R1 to R4, Fig. 2 and Supplementary Fig. 2; Supplementary Table 2).

The second package is composed of gravels and paleosols, indicating paleochannels and swamp clays associated with a fluvial system (C1-2, C3, C6), characterized by alternating channel and overbank with

**Fig. 2 | Simplified geology** *(upper part of the figure)*, **paleoriver network, and the location of archeological sites.** The beds are slightly tilted towards NE, exposing older rocks towards the southwest and younger rocks in the Namorotukunan Hill in the north. Several linear horizons are characterized by coarse clastic sediments (e.g., cobbles, pebbles, and sands) with distinct bedding features (e.g., cross-through bedding) representing paleo-channels belonging to paleo-river systems whose flow direction is marked as light blue arrows. A series of geological sections (A-G) *(lower part of the figure)* were used to develop a synthetic log and the magnetostratigraphic context. The main lithologies include red paleosols (R1–R4), sands and poorly cemented sandstones (S1–S3), gray to gray-brown paleosols (P1–P6), conglomerates and gravels (C1–C7), lacustrine clays (L1–L2b), lacustrine sands with ripple marks (LS1), and diatomite-rich lacustrine clays (LD). These lithologies reflect a variety of depositional environments, from fluvial and alluvial systems to lacustrine settings. Green bars reflect straight line distances between stratigraphic sections. Image derived from UAV imagery collected on site.

intervals of paleosols, interpreted as belonging to a floodplain (S3 to C7). All three artifact-bearing horizons are situated within this sedimentary package. The lower part of this interval contains darker paleosols (P0-P3) rich in carbonate concretions and rhizomes, which alternate with sandy gravel deposits that preserve sand casts of plant roots and rhizomes (Supplementary Fig. 2). Brownish-red, poorly developed paleosols characterize the upper part (P4-P6). Some of the upper paleosol levels (e.g., P5) feature large irregular desiccation cracks (Supplementary Fig. 2), filled with well-cemented sands from the overlying bed (e.g., C6). Cut and fill sequences are limited to three local paleo-rivers corresponding to levels C1-2, C3, and C6. The uppermost cut (C6) is the most laterally extensive and shows characteristics of a meandering river. In some areas, this channel eroded underlying sediments down to the paleosol level P3. These cut and fill structures, representing larger paleoriver features, are highly localized and transition laterally into sandy-silty beds interpreted as overbank deposits.

The artifact-bearing horizons occur in the paleoriver system, where currents in the main channel of the ancient, braided river systems were robust enough to transport cobbles and pebble-sized clasts that served as the source materials for the stone tools. The paleoriver system included low-energy environments, where slack water and point-bar deposits as well as floodplain soil surfaces provided the geological context for the preservation of artifacts and faunal remains.

The uppermost and third sedimentary package lies above paleosol level P6 and consists of lake clays (L1 to L2b, see Fig. 2, and Supplementary Fig. 2), occasionally featuring sandy units (e.g., LS1) or diatom-rich sediments (e.g., LD) and a single level containing frequent catfish fossil remains (LS1-L2a transition). Previous investigations conducted on the slopes of Namorotukunan Hill (see Fig. 7C in Baldes et al.[24]) identified the Burgi Unconformity at the base of the L1 bed. However, no age constraints were provided to determine the extent of the time gap represented by this unconformity, leaving its duration, if any, unresolved.

Within the investigated Area 40 sedimentary sequence, four age markers were identified that allowed development of a comprehensive age model for the entire sequence (Fig. 3). The oldest of these markers is a tephra marker bed, which represents the base of the interval. This layer has been identified as Tulu Bor Tuff β[21] and is dated by a $^{40}$Ar/$^{39}$Ar feldspar age of $3.44 \pm 0.02$ million years[22,25–27]. Geochemical analyses (Supplementary Table 1) also confirm this as the Tulu Bor Tuff β.

Paleomagnetic investigations of all the lithologies present, except for the unconsolidated coarse sands (R1–R4), revealed four distinct magnetic polarity intervals (Supplementary Fig. 3). The lowest part of the section, with a thickness of 16.5 meters, corresponds to a normal polarity zone (NZ1). Subsequently, there is a section ~20 meters thick (RZ1) that features reverse polarities. This is followed by a shorter normal polarity segment (NZ2), around 9 meters thick. The top of the Namorotukunan Hill, less than 1 meter thick, is characterized again by the beginning of another reversed polarity zone (RZ2). In most of Area 40 this unit has been removed by modern erosion.

The Tulu Bor Tuff forms the lower boundary of the sequence at 3.44 Ma, and the Paleo-Lake Lorenyang transgression[28,29] marks its upper boundary at around 2.2 Ma. These two markers allow us to correlate the local magnetic polarity sequence observed at Namorotukunan with the established Geomagnetic Polarity Timescale

(Supplementary Data 1, Supplementary Data 2 GPTS[30]). Specifically, NZ1 corresponds with the Gauss (C2An) Chron, RZ1, and RZ2 represent parts of the Matuyama (C2r) Chron, and NZ2 correlates with the Feni excursion (Formerly Réunion excursion)[31] or, alternatively, with the Olduvai sub-chron. The Mammoth (C2An.2r) and Kaena (C2An.1r) reverse polarity subchrons from the lower part of the Gauss Chron were not found. These are either located in the sandy intervals S1 and S2, (incompatible with paleomagnetic analysis) or these sandy intervals are associated with disconformities. This limitation impacts our estimations of sedimentation rates in the lower interval but critically, does not significantly affect the age estimation of the artifact levels investigated here.

The studied interval includes the Upper Burgi Unconformity (UBU), a surface interpreted in parts of the Koobi Fora Formation as marking a depositional hiatus of several hundred thousand years[24]. However, the UBU is not clearly expressed in Area 40, where the transition from fluvial to lacustrine facies is conformable and no field evidence for an erosional surface is present. The overlying lake clays match Upper Burgi lithologies described by Baldes[24] and the absence of the KBS Tuff in these well-preserved sediments—despite its occurrence ~600 m west of Namorotukunan across the North Gele Fault[21]—suggests that the Namorotukunan section does not reach the KBS Member. If a hiatus is present, it must be confined to a narrow interval within RZ1, implying a much shorter duration than has been proposed for central Koobi Fora. We thus consider two possibilities: Scenario A, in which sedimentation is continuous, and Scenario B, in which the UBU represents a brief interruption. Both scenarios securely place the archeological levels in the late Pliocene–early Pleistocene and do not alter our behavioral or chronological interpretations.

Considering the correlation of R1 with the Matuyama chron, we are presented with two correlation options for the NZ2 polarity interval: either aligning it with the Feni excursion or the Olduvai sub-chron. We favor the first option—the correlation with the Feni excursion—due to the absence of the KBS Tuff in the lacustrine sediments corresponding to the NZ2 polarity zone. The KBS Tuff is an important and geographically expansive stratigraphic marker that, if present, would support a correlation with the Olduvai subchron. KBS has been described ~600 west of Namorotukunan, across the North Gele fault[21], overlaying a thick succession of lake clays, inconsistent with the section at Namorotukunan. Its absence in Namorotukunan suggests that the NZ2–RZ2 interval more likely corresponds to the Feni excursion.

In addition to the level of the Tulu Bor Tuff β ($3.44 \pm 0.02$ Ma), paleomagnetic data provide three new chronological points for developing the age model for the studied section and obtaining an age estimate for the artifact-bearing horizons at Namorotukunan (Fig. 3). These points are the Gauss-Matuyama reversal (NZ1/RZ1) at 2.610 million years[30], the onset of the Feni subchron (RZ1/NZ2) at 2.137 million years, and the termination of the Feni subchron (NZ2/RZ2) at 2.116 million years[32].

We present two alternative age models based on the different correlations described above. These age models reflect two possible scenarios: the first scenario (Fig. 3a) assumes continuous sedimentation between gravel levels C7 and lake clays L1, while the second scenario (Fig. 3b) incorporates the Burgi Unconformity at the C7/L1 lithologic boundary, as previously suggested by Kidney[21] and Baldes

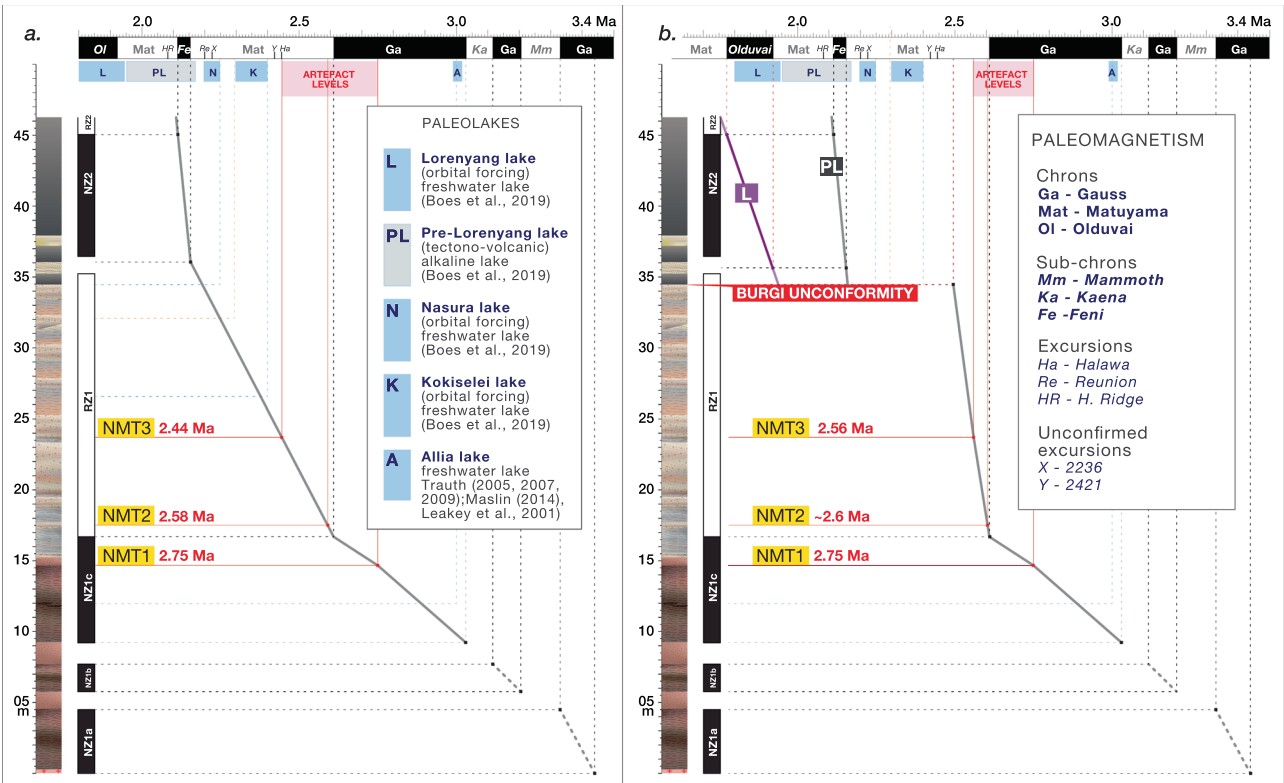

**Fig. 3 | Age-depth models for Namorotukunan.** obtained by correlating the depth axis, represented by the polarity pattern obtained from Namorotukunan (*vertical axis*) with the age axis, represented by the Geomagnetic Polarity Timescale[99] combined with additional geochronological data (*horizontal axis*), represented by paleomagnetic data represented by magnetic chrons, subchrons and excursions[31] and lake phases described from the Turkana Basin[28,47,100,101]. Two scenarios are considered: a. scenario a assumes continuous sedimentation between gravel levels C7 and lake clays L1; b., represents the second scenario which places the Burgi Unconformity at the C7/L1 lithologic boundary, as previously suggested by Kidney[21] and Baldes et al.[24], providing slightly older estimates for sites NMT2 and NMT3. The scenario with the Burgi Unconformity provides two correlation options for the NZ2-RZ2 polarity interval, either with the Feni excursion (PL) or the Olduvai sub-chron (L). We favor the first option due to the absence of the KBS tuff in the lacustrine sediments corresponding with the NZ2 polarity zone. For lithological column and paleomagnetic polarity zone legend, see Fig. 2.

et al.[24]. This second scenario results in slightly older age estimates for sites Namorotukunan-3 (NMT3) and Namorotukunan-2 (NMT2).

In this paper, we use the first scenario, which provides more conservative (younger) age estimates. It is important to note that these scenarios do not affect the age estimation of the oldest archeological levels presented in this paper.

## Archeological findings

Archeological research was carried out within the paleontological collection Area 40 at the Namorotukunan site (36.329 E, 4.399 N). Three excavations (in a ~ 5000m² area), named Namorotukunan-1 (NMT1), Namorotukunan-2 (NMT2), and Namorotukunan-3(NMT3), corresponding to discrete lithological units S3-C1 (for NMT1), C2-P2 (for NMT2), and C3 (for NMT3) yielded a total of 1290 artifacts: NMT1 (n = 198), NMT2 (n = 775), and NMT3 (n = 317), along with associated fossils (Supplementary Table 3). Artifact-bearing horizons are concentrated in sands and fine gravels (Fig. 4). Artifacts associated with the C3 horizon, (NMT3) appear to have undergone minor post-depositional disturbance as indicated by the absence of the smaller fraction of artifacts and linear orientation for this 2.44 Ma archeological horizon (NMT3; see Supplementary Figs. 4, 5, 6,7). Abrasion on artifacts is present in some of the artifacts, yet thin sharp edges are preserved in the assemblages (% of assemblage exhibiting abrasion ranges from 18-22%). Geochemical and mechanical damage to the surfaces of many of the fossil bone specimens prevents a detailed taphonomic analysis of bone modification in the faunal assemblage. However, some specimens within the 2.58 Ma assemblage include well-preserved surfaces that possess butchery marks indicating that hominins used tools to extract high-quality dietary resources from large mammalian remains (see Supplementary Fig. 8).

## Detailed Descriptions Of The Archeological Finds

Artifacts from Namorotukunan exhibit many features that are diagnostic of anthropogenic conchoidal fracture (e.g., prominent bulbs of percussion, clear striking platforms, contiguous flake scars; although see Proffitt et al.[9,33]) and are dominated by smaller flakes and simple cores. High proportions of sharp-edged flakes and fragments, comprising 94.2 to 79.4% of the assemblages, indicate sharp edges were the likely focus of this technology. Early Oldowan assemblages tend to have flakes that are of similar dimension to their cores (as opposed to Acheulean sites where cores are much larger) and have fewer flake scars on cores (as compared to younger Oldowan assemblages[3,12] (Fig. 5). Various technological attributes indicate that the Namorotukunan assemblages are more similar to other early Oldowan assemblages (e.g., Bokol Dora 1; Nyayanga) than assemblages younger than 1.8 Ma (Fig. 5; and Supplementary Fig. 10,11). Evidence of an understanding of fracture mechanics (e.g., relationship between platform angles and other technological features) is similar between hominins that produced the assemblages at Namorotukunan and older Oldowan assemblages (Supplementary Fig. 11). Cores from the Namorotukunan assemblages also show similar patterns to those seen at other early Oldowan assemblages ( > 2.3 Ma) where cores are rarely rotated during production (% Cores that are Unifacial; Fig. 5). Although percussive tools are infrequent in the Namorotukunan assemblage, the

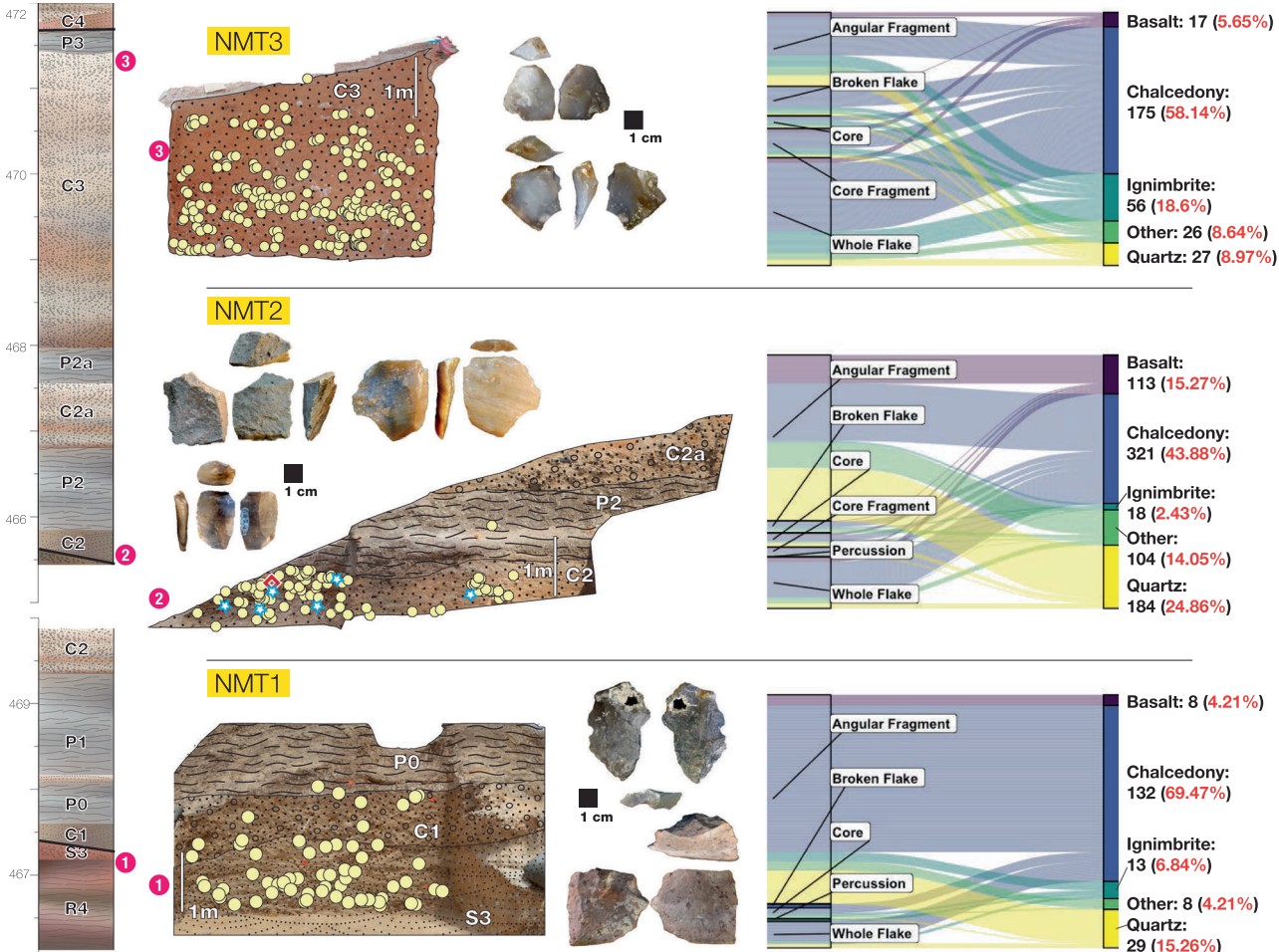

**Fig. 4 | Wall sections of three main stratigraphic units.** The three archeological horizons (NMT3 dated to 2.44 Ma; NMT2 attributed to 2.60 Ma; and NMT1 attributed to 2.75 Ma). These archeological horizons are correlated to the composite stratigraphy for the Area 40 region. Wall sections depict the distribution of artifacts in these stratigraphic units. The right diagrams reflect the proportion of artifacts in major typological categories[102] and the major rock types present in these archeological assemblages (see Supplementary Data 3). A high prevalence of chalcedony is evident in all the archeological horizons. Yellow circles represent individual stone artifacts greater than 2 cm, white-blue stars represent faunal specimens recovered in situ during excavations (see Supplementary Fig. 6), and the red diamond represents the cut-mark bone described in Supplementary Fig. 8. Note that fauna was poorly preserved in the older and younger time horizons.

appearance of heavily battered artifacts (<1% of assemblage; NMT2) confirms the presence of percussive activities (Supplementary Fig. 12). Comparisons of raw material selectivity demonstrate fine grained chalcedony at proportions that are significantly greater than its presence in nearby conglomerates (Supplementary Fig. 13). We note that although hominins at Namorotukunan appear to be selective in their selection of specific rock types, there is little evidence of transport of stone. Gravels (braided river system deposits) that carried cobbles suitable for artifact manufacture were present in coarser grained sediments directly adjacent to all archeological sites. This level of selectivity is evident in all three assemblages and suggests a persistent preference for materials that fracture consistently (Supplementary Fig. 14).

### Paleoenvironmental proxies

The stratigraphic sequence in Area 40 of the KF Fm provides a framework for examining the paleoecological conditions from ~3.44 to ~2.0 million years ago. The existence of paleosols, along with accompanying sediments and fauna, facilitates a multiproxy reconstruction of the fluctuating environmental conditions within this period. Pedogenic carbonates, plant wax biomarkers, and phytoliths provide detailed information (Fig. 6, and Supplementary Fig 15 and Supplementary Data 4) on the physiognomic structure of the

ecosystems across the late Pliocene and early Pleistocene in the region. The range of values from some paleoenvironmental proxies indicate the presence of diverse habitats within some temporal intervals (e.g., $\delta^{13}C$ values of pedogenic carbonates −10 to −2 ‰). However, when fitting linear estimates through the data, there are coincident changes that suggest an increase in C4 vegetation and a decrease in water availability in the gravel and paleosol interval (2.7-2.2 Ma; Supplementary Fig. 16). This is supported by increases in the $\delta^{13}C$ values of C31 and C33 $n$-alkanes that indicate an increase in $C_4$ vegetation and likely, open habitats, in the gravel and paleosol interval.

Phytolith analysis indicates an overall decrease in grass phytoliths that characterize the gravel and paleosol interval (Fig. 6). High incidences of palm and sedge phytoliths before these red silts, paleosols, and sands interval, indicate the presence of substantial standing water in these habitats (3.4 – 2.6 Ma) that is notably absent in the younger sediments, the gravel and paleosol interval (2.6-2.2 Ma). Significant increases in microcharcoals, associated with landscape-scale wildfires, point to a more seasonally arid habitat beginning around the onset of the gravel and paleosol interval (2.7-2.6 Ma; Fig. 6). Seasonality probably played a key role, as shown by investigations of paleo-fire frequency on more recent timescales in Africa[34]. Phytoliths from the 2.7-2.2 Ma time interval (Fig. 6) show

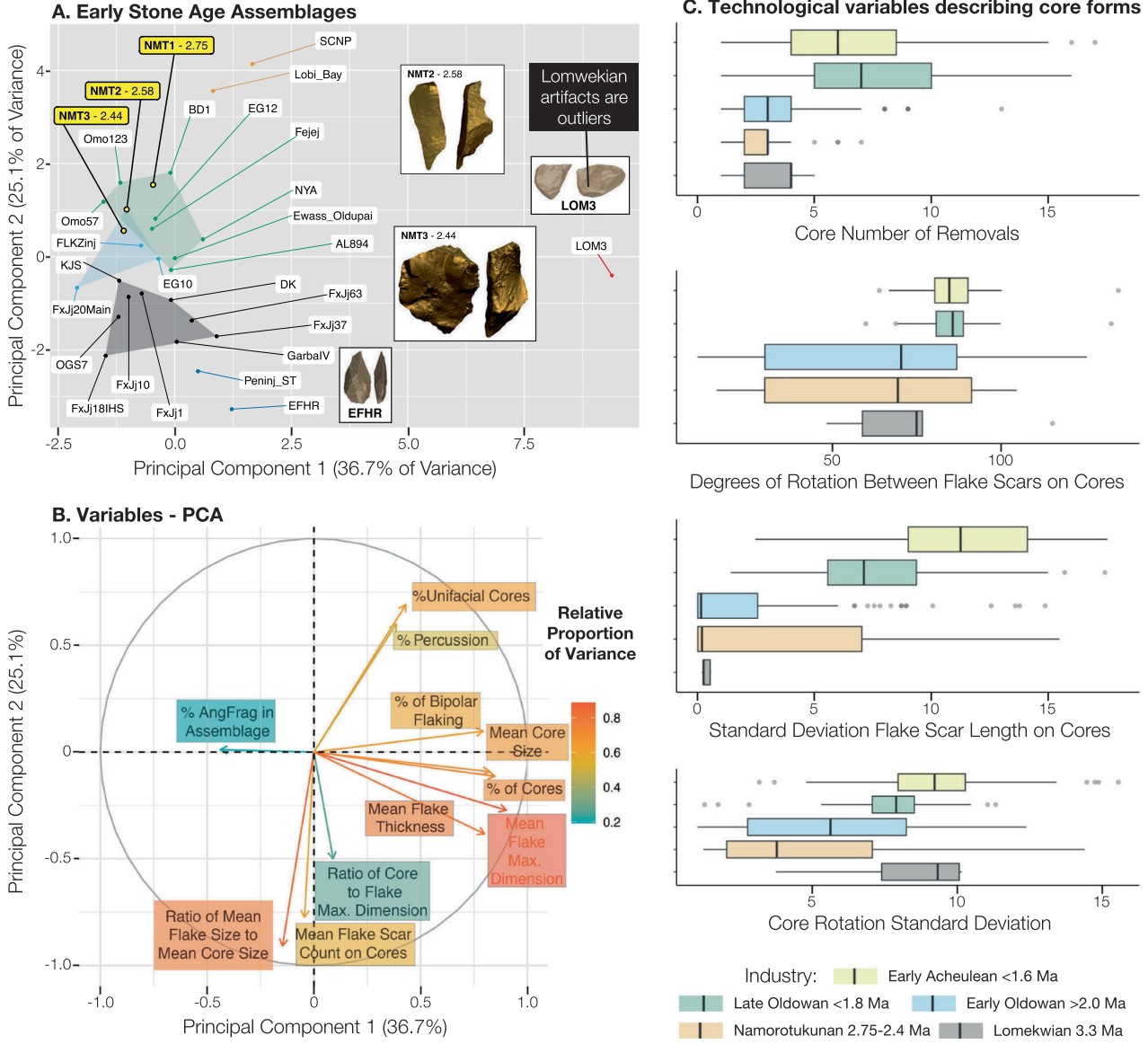

**Fig. 5 | Early Oldowan technology. A** Principal components analysis of major technological features of several of the earliest stone tool industries currently known (Supplementary Data 6, 7). PCA scores were used to calculate a K-means cluster analysis. The shaded polygons (convex hulls) are the results of this K-means cluster analysis. In addition to early archeological assemblages, we include here assemblages made unintentionally by both Capuchins (SCNP[33]) and long-tailed macaques (Lobi Bay[9]). **B** The analysis of eigenvectors indicates that most Oldowan (and primate) assemblages are distinguished from Lomekwi by their smaller size and relative infrequency of cores. Most early Oldowan assemblages cluster together based on the relative size of flakes compared to cores. **C** Technological variables describing core forms analyzed from 3D models of cores (See "Methods"). Boxplots represent the interquartile range of data. Whiskers represent the upper and lower quartile.; In numerous variables the Namorotukunan assemblage is similar to early Oldowan ( > 2.0 Ma) assemblages but significantly different from the Lomekwi 3 assemblage as well as the later Oldowan and early Acheulean assemblages (Supplementary Figs. 9, 10, 11).

---

increased shrub-derived morphotypes and decreased grass phytoliths, further suggesting a shift towards more seasonally arid habitats. This is further supported by the presence of phytoliths of C4 Chloridoideae grasses suggesting xerophytic habitats consistent with arid conditions at this time.

Chemical index of alteration (CIA) and mean annual precipitation (MAP) was calculated based on bulk geochemistry of paleosol samples (see "Methods" and Supplementary Note 2). High CIA values indicate removal of labile cations ($Ca^{2+}$, $Na^+$ and $K^+$) relative to $Al^{3+}$ during the chemical weathering of silicate minerals under wet climatic conditions and low values indicate absence of the chemical weathering under dry climate[35,36]. CIA values (51 to 64) indicate incipient to moderately weathered paleosols. Low CIA values (<55) between 2.9 and 2.7 Ma indicates dry climate (Fig. 6). MAP calculated using geochemical climofunctions for paleosols varies between 231 and 855 mm/year, broadly classifying the climate as semi-arid to semi-humid. MAP follows a similar trend as of CIA and lowest MAP is recorded between 2.7 and 2.6 Ma indicating an increased aridity at this time interval. The time frame between 2.7 and 2.2 Ma is associated with the relatively low MAP estimates. This pattern is reversed at the return of lacustrine environments (i.e., Paleo-Lake Lorenyang) at the top of the sequence (2.2-2.0 Ma) (Fig. 6).

Environmental magnetic analysis of bulk sediments confirms the reliability of our paleomagnetic approach. The analysis indicates that magnetite is the primary iron oxide, carrying magnetic properties in nearly all samples. However, at the onset of the gravel and paleosol interval (Supplementary Fig. 17), we detected a notable exception: evidence of a second iron mineral, hematite.

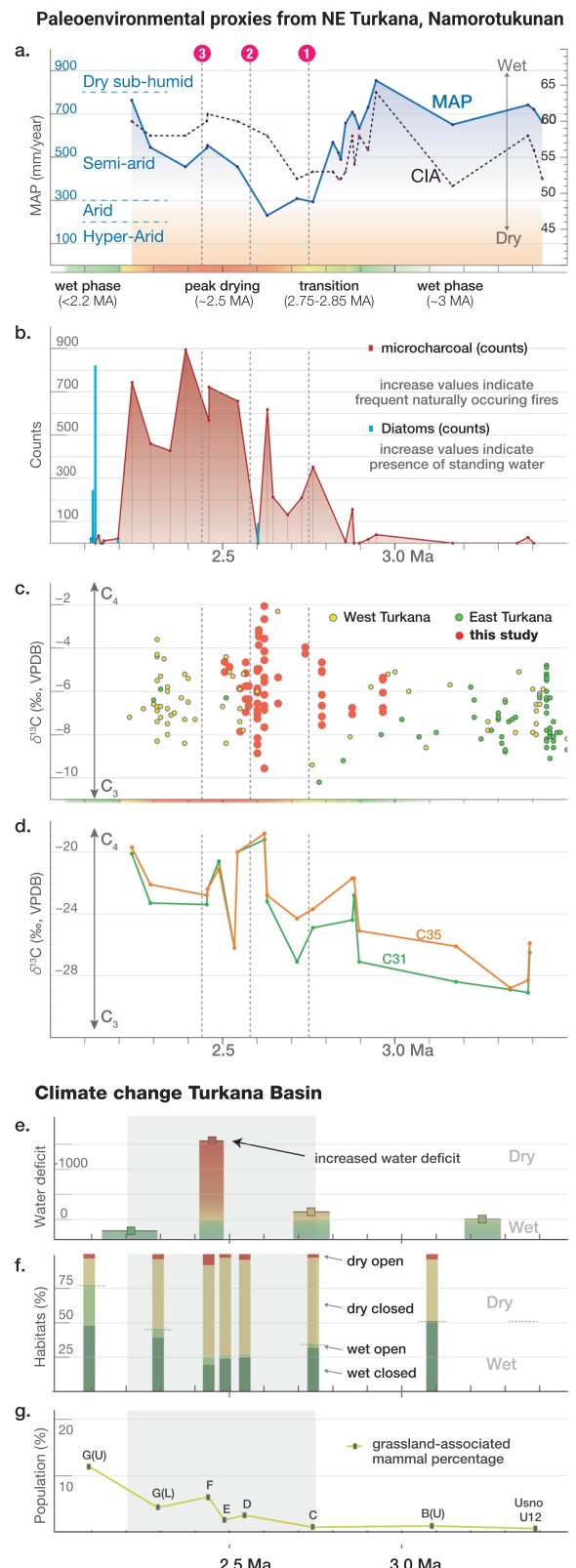

Paleoenvironmental proxies from NE Turkana, Namorotukunan

Climate change Turkana Basin

**Fig. 6 | Paleoenvironmental reconstructions. (Local Scale) Paleoenvironmental change, based on proxy data obtained in this study. a** Mean annual precipitation (MAP) estimates obtained from paleosol geochemistry (mm/year) follow a similar trend with the Chemical index of alteration (CIA), red numbered dots represent the three archeological horizons (1. NMT1, 2. NMT2, 3. NMT3); **b** Microcharcoal and diatoms found in the phytolith samples (counts); **c** Composite record of pedogenic carbonate $\delta^{13}Cpedcarb$ values from East and West Turkana, compiled from existing literature by Levin (2013) and combined with new paleosol $\delta^{13}Cpedcarb$ data from site Namorotukunan and its surroundings (‰, VPDB); **d** An overview of paleoclimatic trends observed from plant wax biomarkers in Namorotukunan, in this study. **(Basin Scale) Climate change in the Turkana Basin: e** Changes in the landscape composition based on faunal composition[48]; **f** Changes in the abundance of open country adapted bovids[90]; **g** Water deficit (ES-EI index) estimates (mm/year) based on $\delta^{18}O_{enamel}$[91,92].

## Discussion

The archeological sites examined in this study were recovered from sediments that range between 2.75 and 2.44 Ma. Deposits representing this time interval, which includes the Burgi Unconformity, were previously thought to be absent in most of the northern portion of the KF Fm[21,24,39]. However, sediments of this age have been documented in various locations within the northern Koobi Fora Formation[40]. The presence of dated horizons confirms active sedimentation on the east side of Lake Turkana in the terminal Pliocene [(e.g., Aberegaiye Tuff (2.7 Ma) in Area 116[40] and Area 130[21]; Tuff D-5-2 of the Shungura Fm. (~2.44 Ma) in Area 139[21]]. The oldest archeological horizon (NMT1) is situated in unit C1-S3 and lies 2.3 meters below the Gauss-Matuyama boundary (C2An-C2r), with an attributed age of 2.75 million years. Excavations in unit P2/C2 (NMT2) directly overlie the Gauss-Matuyama boundary and are estimated at 2.60 million years. The uppermost archeological excavation is located in unit C3 and is positioned stratigraphically ~2 meters above the Gauss-Matuyama boundary (C2An-C2r), with an attributed age of 2.44 million years. This aligns with previously reported known Oldowan assemblages which also appear in the upper Gauss Chron (Bokol Dora 1[12]; Nyayanga 3[3]).

Hominins at Namorotukunan appear to have had an acute understanding of the mechanical properties of rocks used for artifact manufacture. Numerous lithologies were available to make stone artifacts (including basalt, which was overwhelmingly favored for tool production later in the Koobi Fora Formation sequence [Supplementary Fig. 13[41]], yet the degree of selection for fine-grained raw materials at the assemblages at Namorotukunan is similar to or exceeds levels of selectivity in younger assemblages (Supplementary Fig. 13). Fine-grained crypto-crystalline rocks, (e.g., chalcedony, jasper) available in conglomerates in the Koobi Fora Formation, have mechanical properties that result in predictable fracture patterns (Supplementary Fig. 14[42]). This selection for fine-grained raw materials is a characteristic feature of Oldowan assemblages (Supplementary Fig. 13). Artifacts from Namorotukunan indicate that hominin toolmakers had a clear understanding of some components of stone knapping as indicated by the interaction between flaking angles and where they struck cores (Supplementary Fig. 11). This is a substantial departure from flakes produced by percussive activities (Supplementary Fig 18). Whole flakes produced in each assemblage from Namorotukunan indicate morphological overlap with individual flakes produced during percussive tool use by modern non-human primates. However, at an assemblage scale, flakes from Namorotukunan are distinguished from those produced by percussive activities by their shapes and technological attributes (Supplementary Fig. 18[9], Supplementary Data 8). Analysis of the relationships between flaking surfaces indicates that Namorotukunan hominins rotated cores less frequently than those found in later Oldowan assemblages[43] (Fig. 5), suggesting that hominins had not mastered some of the features associated with advanced flintknapping skills[44].

There are relatively few identifiable faunal specimens from the entire sedimentary sequence in Area 40 (Supplementary Data 5). However, those species that are present in the gravel and paleosol interval indicate relatively open habitats. Antilopin and alcelaphin bovids as well as equids are the most abundant taxa. Several suids are also present, but most notably those associated with grass dominated habitats (e.g., *Metridiochoerus*[37,38]).

In many respects the overall technological pattern of all the Namorotukunan assemblages most closely aligns with the known Oldowan sites (Fig. 5, and Supplementary Figs. 9, 19). The older, 3.3 Ma Lomekwi 3 assemblage represents a distinctive technology compared to Namorotukunan and other early Oldowan sites (Fig. 5). Lomekwi 3 includes extraordinarily large cores, with relatively few flaking surfaces and extensive evidence of battering associated with percussive activities[6]. The data from Namorotukunan further supports previous analysis that indicate that the earliest Oldowan represents a distinct technological strategy from that which is seen in the earlier Lomekwian technology[3,12]. Previous analyses of the timing of the Oldowan record suggest the temporal gap between the Lomekwian and Oldowan is expected to narrow with further discoveries. This assertion is supported by the expansion of Oldowan assemblages earlier in time[45].

## Landscape and climatic shifts

Paleoecological information from the northeastern portion of the Koobi Fora Formation, from the Tulu Bor Tuff β (3.44 ± 0.02 Ma) through the Paleo-Lake Lorenyang transgression in the Turkana basin -2.2 million years ago[28,29], indicates major ecological shifts associated with the earliest appearance of stone tool-using hominins. Paleoenvironmental reconstructions from older records[46], in the vicinity of Namorotukunan (~25 km. to the south in Area 129) suggest high variability between $C_4$ and $C_3$-dominated ecosystems ~3.60-3.44 million years ago. Directly after the Tulu Bor volcanic eruption, ~3.44 million years ago, the paleo landscape around Namorotukunan exhibited characteristics of a humid floodplain situated near a permanent body of water. High occurrences of palms and sedges before 2.8 million years ago suggest the existence of significant standing water or a high water table in these environments (Fig. 6). MAP is estimated to have reached around 855 mm per year in the older sediments (Fig. 6), marking one of the highest rainfall values documented within the Area 40 sequence. Around 2.8 to 2.7 million years ago, the landscape underwent a profound transformation, transitioning into alternating channel and floodplain deposits (lithofacies) of a paleoriver system. This change likely resulted from a retreat of the paleolake shoreline due to the onset of a drier climate, as indicated by the decrease in MAP to under 300 mm per year (Fig. 6), as reflected in paleosol geochemistry at Namorotukunan after 2.75 million years ago. Paleo-vegetation studies (e.g., phytoliths, plant-wax biomarkers) provide further support for the hypothesis of a vegetation reorganization after 2.75 million years.

The older part of the stratigraphic sequence ( > 2.75 Ma) is characterized by a higher occurrence of $C_3$ vegetation (Fig. 6). An increase in $C_4$ vegetation occurs after 2.75 million years. The general increase in $\delta^{13}C$ values observed in the $n$-alkanes reflects a period of perturbation, with significant fluctuations, likely triggered by changes in vegetation and adaptation to drier climates (Fig. 6). Phytolith analysis indicates an overall reduction in grass phytoliths around 2.9 million years ago (Supplementary Fig. 15). The presence of phytoliths indicating shrubs and a notable increase in microcharcoals (evidence of landscape-scale wildfires) point to a more arid environment around 2.7–2.8 million years ago. This is further reinforced by the presence of C4 Chloridoideae grasses, suggesting xerophytic habitats consistent with arid conditions after 2.8 million years ago.

~2.2 million years ago, evidence from facies changes at Namorotukunan and in the nearby ( < 200 m laterally) sediments suggest that the landscape experienced yet another transformation to a lacustrine domain[28]. At this time much of Area 40 was flooded by the Paleo-Lake Lorenyang, a precursor to the modern-day Lake Turkana[28]. Following an initial inundation, that left the Namorotukunan area completely submerged, the lake receded, and the area around Namorotukunan evolved into a shallow-water nearshore depositional environment. However, this regression was short-lived, and the lake expanded once again. The base of this subsequent lake transgression is characterized

by rich catfish bone beds that may be indicative of an environmental catastrophe within the lake. This has previously been documented in West Turkana, resulting from a shift to an endorheic regime following the closure of the lake's outflow[47]. During the late Pliocene, the EARS witnessed significant landscape and climate transformations such as lake regression and aridification (Fig. 6; e.g[15,48–54].) that significantly transformed vegetation communities, expanded grasslands, reduced woody cover, and altered the physiognomic composition as well as available resources to hominin and non-hominin species[55]. These changes were instrumental in shaping the evolution and adaptations of both flora and fauna in eastern Africa as we show locally in this study[15,46,51,56,57]. There is a connection between local and regional environmental shifts and the appearance of the Oldowan tool technology on the eastern side of the Turkana Basin. Artifact levels represent records of localized paleolandscapes shaped by a unique convergence of conditions. Geological factors ensured the availability of large cobbles that can be used to make stone artifacts, while features of paleoriver systems (e.g., point bars) facilitated the rapid burial of artifacts in archeological deposits. Paleoecological conditions provided foraging resources that encouraged tool use (i.e., open habitats had higher frequencies of resources that required tool use[58]). The co-occurrence of hominins as well as local ecological conditions that favored tool manufacture and burial resulted in the presence of these concentrations of artifacts. These conditions were met during a relatively dry climate episodes in the Namorotukunan area between 2.75 and 2.44 Ma. Factors that favored the production and burial of artifacts likely shifted within the basin in concert with global and regional environmental factors.

Orbital variability has played a major role in shaping the climate and landscapes in eastern Africa, including during the late Pliocene and Pleistocene interval studied here, primarily through its influence on the intensity of the East African monsoon[59,60]. A pivotal event during this period was the closure of the Panama Seaway roughly between 2.8 to 2.5 million years ago, which had profound implications for Earth's climate and ecosystems[61]. This closure disrupted ocean circulation patterns and contributed to the formation of the Atlantic Ocean Gulf Stream, exerting a substantial influence on North America and Northwestern Europe[62]. The reconfiguration of the paleogeography brought about changes in sea surface temperatures in the Pacific, affecting both the Western Pacific and the Americas[63]. This led to a cascade of climatic effects, including the rapid strengthening of the Westerlies at around 2.75 million years ago[64] and a similar strengthening of the NE trade winds at about 2.7 million years ago, as documented through various records offshore of western Africa[65]. This shift in wind directions and their overall influence on aridification in various parts of Africa likely influenced the overall drying trend, the increase in C4 vegetation, and possibly increased wildfires based on microcharcoal data presented here.

In the context of human evolution, the late Pliocene climate and ecosystem changes in eastern Africa likely exerted selective pressures on all fauna, including hominins. It influenced their dietary adaptations and behavioral strategies as they adapted to shifting ecological pressures[66]. As these shifts impacted different parts of Africa separately, it is likely that such selective pressures on tool use impacted different populations at different times[54]. This period of aridification played a role in shaping the ecological context in which the behaviors of our human ancestors evolved[67].

## Resources, Incentives, and Opportunities for Stone Tool Technology

Presently the early Oldowan technology is assigned to the Late Pliocene and coincides with substantial environmental transformations documented across eastern Africa at the close of the Pliocene[15,48,51,68]. The significant climatic shifts seen throughout eastern Africa ~2.75 Ma, highlighted in the sequence at Namorotukunan, may have heightened

the need for hominins to expand their dietary niche by incorporating high-quality dietary resources (i.e., animal tissues) into their diet[14,66,69]. This shift in the dietary niche resulted in a subsequent release of selective pressures on the various hominins living on this landscape, including the earliest known members of the genus *Homo*[70] and occurs concurrently with environmental shifts in Ethiopia[15,55]. Elsewhere in eastern Africa these changes are associated with the appearance of early *Homo*[11,15,55]. The hominin fossil record in the KF Formation at this time is limited. Further south in the Koobi Fora Formation, KNM-ER 5431 has been attributed to early *Homo*[71]; but also shares affinities with *Australopithecus*[72]) and may be older than Namorotukunan[73]. Currently, detailed chronostratigraphic information for these specimens is not well resolved[74].

The development of consistent production of sharp-edged tools likely improved the foraging energetic return of hominins in the Pliocene[16], similar to the increased foraging efficiency seen in various extant primates[75]. Controversial evidence of the earliest technology elsewhere in eastern Africa at Lomekwi[6] (although see Archer et al.[7]), along with the butchered bones from Dikika, Ethiopia[76] (although see Dominguez-Rodrigo[77]), indicate that tool use could be a shared ancestral trait (i.e., plesiomorphic), already present in the last common ancestor with *Pan*, and may reflect a more generalized primate pattern of tool use[9,78].

The prevalence of the Namorotukunan artifacts in multiple stratigraphic levels is linked to the frequency of particular paleogeographic features that record the presence of paleo-river systems. Importantly, even though the different stratigraphic horizons are separated by hundreds of thousands of years, the regular association between such river systems and stone artifacts is ubiquitous in the sedimentary record at Namorotukunan. As with similar localities in eastern Africa the presence of rivers that transported and provided raw material appears to structure the frequency of stone artifact production[41,79]. In the northeastern portion of the Turkana Basin, large rivers transporting raw material are intermittent in the sedimentary sequence between 3.4 and 2.1 Ma (Fig. 2). River systems that carry cobbles only appear ~2.7 Ma, in Area 40, coincident with the oldest archeological levels. This suggests that this technology in the Namorotukunan area was likely both limited and facilitated by raw material availability[5]. Therefore, the absence of stone tools earlier in the record of this region could indicate fluvial source area and drainage paleogeography that result in a paucity of rocks available to make stone artifacts.

Our paleoenvironmental data reveal climate-driven dynamic landscape evolution in East Turkana during the late Pliocene. This period witnessed the disappearance of lake-proximal environments after 3.44 Ma, which were replaced by a semi-arid alluvial landscape with episodes of paleoriver activity including channel and floodplain (paleosol) deposition[28,39,52] until the region experienced a re-inundation by the Paleo-Lake Lorenyang around 2.2 million years ago. These paleo landscape transformations likely mirror fluctuations at regional and global scales. Increased aridity from ~2.75 million years ago was followed by a transition to more humid conditions around 2.1 million years ago (Supplementary Fig. 22). The large assemblage of flaked stone artifacts from Namorotukunan indicates that the onset of the production of sharp edges in the Namorotukunan region is associated with these environmental changes in the Koobi Fora Formation. Evidence for similar shifts near the Plio-Pleistocene boundary has been documented in multiple parts of the EARS[15,55,68].

Comparable to other early Oldowan sites (e.g., Bokol Dora, Gona: Ethiopia and Nyayanga: Kenya), the emergence of regular production of sharp flakes at the Namorotukunan site represents a derived trait complex, possibly connected to innovative foraging behaviors[3,12]. The presence of butchery marks on one specimen (Supplementary Fig. 8; NMT2) confirms that a portion of this adaptation was associated with access to some large mammal tissues from carcasses[14]. A possible focus on meat from carcasses is further supported by the prevalent use of crypto-crystalline silica in the artifact assemblages. This type of rock, known for its uniformity and high silica content, is excellent for the production of numerous smaller sharp-edged tools[80], but its appearance in mostly smaller, brittle cobbles makes it unsuitable for percussive tools. The use of sharp-edged tools significantly enhances the extraction of nutritionally rich food sources from mammal tissues[16]. This change in behavior may not align with the appearance of the genus *Homo*, however variability in the available sedimentary record limits our ability to resolve the co-occurrence of biological and behavioral shifts[81].

Oldowan technology in the Turkana Basin appeared during periods of dynamic climate and ecological change. However, the persistent nature of stone tool technology from 2.75 throughout to 2.44 Ma is consistent with the hypothesis that it reflects an enduring adaptation. The extent to which this technology connects to earlier industries or to a more generalized tool use remains uncertain[9,78,82]. The technological innovations that distinguish our lineage, particularly those that emerged in the later stages of the Pleistocene[83], could represent a distinct development that originated independently from the array of varied tool-based behaviors that began in the Pliocene[6,76].

## Methods

### Fieldwork

Excavations at Namorotukunan were conducted in Area 40 (East Turkana Basin, Northern Kenya) between 2013 and 2022. The objective was to excavate known Early Pleistocene localities[19] to provide a multi-disciplinary context for the Oldowan lithic assemblage found at the site (Fig. 1). All material was excavated according to discrete stratigraphic horizons. Archeological material was mapped to the nearest mm using a total station Leica Builder 505 with a TDS Nomad 900 LE using EDM Mobile 5.1bt[84]. Detailed stratigraphic information for many artifacts was captured using three-dimensional photogrammetric models of artifacts in their in situ context. Excavations were conducted under the auspices of a permit provided by the National Council for Science and Technology of Kenya. Excavations and collections were carried out under the auspices of an Exploration and Excavation Permit provided by the Kenyan Ministry of Heritage, Tourism and Culture. This research was carried out with permissions granted by the National Museums of Kenya and the Ministry of Education, Science, and Technology of the Republic of Kenya (MOEST 13/001/31 C 216 and NACOSTI/P/23/31328).

### Artifact and faunal analyses

All artifacts were measured using standard caliper and goniometer measures at the Archeology Division of the National Museums of Kenya in Nairobi. Comparisons of technological variables were compiled from the literature following recent reviews[3]. Analysis of core forms (Fig. 5) included 275 three dimensional models. These models were captured using a structured light scanner (Einscan). Three dimensional models were captured at the National Museums of Kenya (with the exception of the Lomekwi 3 collection which was captured from models available at http:africanfossils.org). Three dimensional models were analyzed using morphometric packages in the R statistical computer language (Morpho, bezier, Rvcg, rgl). Placement of landmarks on three dimensional models was conducted using Mesh-Lab software (2022.10). Variables (e.g., rotation angle between flake scars) were calculated from three dimensional models. Data in this analysis deviate significantly from a normal distribution. As such we calculated differences between groups using a permutation test that compared differences between groups to 10000 resampled simulated assemblages of similar size samples from the entire data set. Significance values are estimated based on the probability of finding similar differences in resampled data sets. Faunal specimens were analyzed in the Archeology Division of the National Museums of

Kenya. Modifications to bone surfaces were identified using a 5X hand lens. Further imaging of modifications took place on a Sensofar Nanofocus spinning disk microscope at the Max Planck Institute for Evolutionary Anthropology.

## Geological description and stratigraphy

Lithological characterization was conducted in situ across various studied sections, where each section was thoroughly examined to identify and describe the rock types present. Observations focused on key characteristics including texture, color, mineral composition, and structural features (Supplementary Table 2). Systematic sampling was performed, with rock samples collected for detailed laboratory analyses.

To synthesize the geological attributes of the area, seven distinct sections (A-G) were logged, sampled, and measured using a combination of geological field techniques and geochemical methods as well as site fabric analysis. Bedding planes and fault planes were measured using a Brunton compass. The overlapping seven sections were combined in a composite section (Supplementary Fig. 3). These descriptions were then integrated to construct a composite synthetic section that represents the overall geological framework and variability across the study area, facilitating a comprehensive understanding of the geological context and its spatial heterogeneity.

Lateral geological features were documented to assess the *horizontal continuity and distribution of rock formations*. These observations were crucial for understanding the lateral extent of geological phenomena and their implications on the regional geological setting. High-detail examination of *cut and fill features* was undertaken to interpret the processes of erosion and deposition that have influenced the landscape. This included detailed logging of stratigraphic sections to identify sequences of deposition, erosional unconformities, and associated geological processes.

We used paleomagnetism and geochemistry to refine age markers in the section. *160 oriented samples* were specifically collected for paleomagnetic studies. Paleomagnetic studies were mainly focused on magnetostratigraphy, aimed at identifying magnetic polarity reversals. Geochemical methods were employed to verify if the signatures of the tephra correspond to well dated tephra in the region. Additionally, *222 unoriented samples* were taken for paleoenvironmental reconstruction analyses. We used a multiproxy approach (including Rock-Magnetism, Geochemistry, Phytoliths, etc.) from well dated stratigraphic levels, to describe the paleoenvironmental shifts in the East Turkana environments.

## Paleomagnetism

Paleomagnetic samples from the Namorotukunan site were collected across seven trench sections (A–G) (Supplementary Fig. S16; Supplementary Note 1) during six field campaigns conducted between 2012-2022. Using a 2.5 cm-diameter diamond-coated bit powered by hand operated air pump (necessary because water cooling would destroy unconsolidated samples) samples were oriented with a geological compass and inclinometer. After extraction, the samples underwent progressive demagnetization in the laboratory. In the Utrecht paleomagnetic facility 160 samples (Supplementary Fig. S18) underwent dual demagnetization techniques: stepwise alternating-field (AF) demagnetization (0–80 mT) and thermal demagnetization (20–600 °C). These methods were employed to isolate primary magnetic components and minimize overprinting.

Samples were measured at the Fort Hoofddijk Paleomagnetic Laboratory (Utrecht University) and at the CENIEH (Centro Nacional de Investigación sobre la Evolución Humana), demagnetization utilized a 2 G Enterprises DC SQUID cryogenic magnetometer with an in-line, triaxial, AF demagnetizer for the AF samples, and an ASC thermal demagnetizer (residual field <20 nT) for the Th samples. The demagnetization results were interpreted using Paleomagnetism.org[85],

Characteristic Remanent Magnetization (ChRM) directions were typically isolated between 15-45 mT or 150-600 °C. Paleomagnetic directions were calculated using principal component analysis[86] based on a minimum of four consecutive steps, ensuring the reliability and robustness of the outcomes (refer to Supplementary Data 1).

For rock magnetic insights, a separate sample set underwent Isothermal Remanent Magnetization (IRM) acquisition curve measurements (up to 1.5 T) using a MicroMag Model 3900 Series VSM vibrating sample magnetometer (Lake Shore Measurements). These rock magnetic analyses complemented the paleomagnetic investigations, providing a comprehensive understanding of the magnetic properties within the sampled materials.

## Tephra analysis

Samples of the Tulu Bor Tuff were collected at (4.399161 N / 36.326657 E WGS 84) and were based on previous descriptions of the Tulu Bor Tuff in Area 40[21]. Samples of the Tulu Bor Tuff were analyzed for geochemistry by electron microprobe (EMP) on a JEOL 8900 Superprobe instrument housed at the Carnegie Institution for Science's Geophysical Laboratory. Samples were measured using wavelength-dispersive x-ray spectroscopy (WDS) using the following analytical setup: accelerating voltage = 12 keV, beam current = 10 nA, and a beam diameter = 10 μm, conditions ideal for reducing alkali loss while obtaining reliable counts for elements such as Fe.

## Pedogenic carbonates

Pedogenic carbonates were collected from a $3 \, km^2$ region within paleontological Area 40. Samples were collected from paleosols with evidence of pedogenesis such as dish structures and root casts. Samples were collected from trenches that were at least 40 cm beneath the estimated top of the pedogenic horizon. Samples were limited to paleosols with discrete nodules. 47 pedogenic carbonates were collected from this region and results are presented in Fig. 6.

Carbonate nodules were isolated from the surrounding sediment using a rotary diamond bit to remove debris, after which they were manually homogenized in a mortar and pestle. The homogenized carbonate reacted with 100% H3PO4 at 90 °C. A dual inlet Thermo MAT253 isotope ratio mass spectrometer analyzed the resulting CO2 gasses for $^{13}C/^{12}C$. Sample results were calibrated using an internal Hagit Carrara marble standard and an NBS-19 calcite standard. The average standard precision over multiple runs was 0.5−0.04 ‰ for $^{13}C$ at Johns Hopkins University.

## Plant wax biomarkers

Sediment samples were collected in the field directly onto Al foil. Samples came from at least 40 cm below modern surfaces and were screened to ensure modern plant material (e.g., roots) was absent. In the lab, samples were rinsed with an organic solvent, dichloromethane (DCM), to remove possible surface contaminants. Samples were then wrapped in foil, placed on a steel plate, and gently disaggregated with a hammer into 1–4 cm size pieces. About 200 g of sample was further crushed into a powder using a solvent-rinsed mortar and pestle. An average of 190.8 g (min: 172.6; max: 198.7 g) of sample was extracted using a Dionex Accelerated Solvent Extractor 350 over three consecutive aliquots (~64 g each) of sample packed into 66 ml cells. Soluble lipids were extracted at 100 °C and 1600 psi (110.3 bar) over four 10-min static cycles with 9:1 DCM:methanol. A recovery standard was added to the total lipid extract (TLE) after extraction. The TLE was dried down under purified nitrogen and compounds were separated into aliphatic, ketone and ester, polar fractions via silica gel flash column chromatography using hexanes, DCM, and methanol, respectively. The aliphatic fraction contained the *n*-alkanes.

*n*-Alkanes were characterized and quantified on an Agilent gas chromatograph (Agilent 7890 A GC) equipped with both a mass selective detector (5975 C MSD) and flame ionization detector (FID).

One microliter of sample dissolved in 100 ml hexane was injected into a multi-mode inlet injector at 60 °C (0.1 min hold), which was then ramped to 320 °C at 900 °C/min and held for the duration of the analysis. Initial GC (gas chromatograph) oven temperature was set at 60 °C (1.5 min hold) and ramped to 150 °C at 15 °C/min, then to 320 °C at 48 C/min. A helium-purged microfluidics device downstream of a DB-5 column (30 m length, 250 mm ID) quantitatively split the GC flow to the MSD and FID detectors. Compound identification was done with comparison of mass spectra and retention times to authentic standards, and quantification was done using MSD peak areas using a response factor correction based upon the peak area of known concentrations of internal standards added to the TLE. The carbon preference index (CPI) was calculated using the equation of Marzi et al.[87] and average chain length (ACL) was calculated as $\sum_{i=a}^{b} iC_i / \sum_{i=a}^{b} C_i$, where $C_i$ is the concentration of the molecule with chain length $i$ and $i = 27$, 29, 31, 33, 35 for n-alkanes.

Carbon isotope ratios of n-alkanes were analyzed using a GC coupled to a Thermo Delta V isotope ratio mass spectrometer through a combustion interface at the Lamont Doherty Earth Observatory Stable Isotope Laboratory. All sample injections were interspersed with injections of molecular mixtures with known isotopic values (mixes A4 and A5 supplied by Arndt Schimmelmann, Univ. of Indiana) that were used for correction of carbon isotope values. Samples were injected in duplicate or triplicate when feasible.

## Phytoliths

To extract phytoliths from sediment, ~1 g of sediment was treated with 10% hydrochloric acid (HCL) to remove the carbonates. Samples were then washed using distilled water and sieved at 250 μm to remove larger particles (>250 μm) and then treated with nitric acid (HNO3) to remove organic matter through an oxidation process. The remaining material was gently sieved through a 53 μm sieve to loosen clay particles adhered to phytoliths for clarity during microscopy analyses. Samples noted to have stubborn clays were further cleaned by repeated sedimentation process and, when necessary, sodium hexametaphosphate was added to deflocculate clays, followed by multiple washing, decanting and centrifuging processes. To float the phytoliths/bio-silica, we added 10 ml of sodium polytungstate with specific gravity of 2.4 g/cm3. Phytoliths were pipetted out to another set of centrifuge tubes for final distilled water washing and recovery.

The recovered residue was dried, and 0.002 g of this was mounted using "Entellen New" mounting media (refractive index 1.53) on a slide and viewed under Carl Zeiss Scope A1 light microscope. Observation, counting and classification was done at 400x magnification. Where necessary, oil immersion was done for Grass Short Silica Cells (GSSCs) morphotypes and photography. Microphotographs were taken by USB 3.0 Digital Camera, THORLABS. Phytolith classification largely followed International Code for Phytolith Nomenclature (ICPN) 1.0 and 2.0[88] for both non-GSSCs and GSSCs phytoliths. Other classification systems were also consulted[89]. Only well preserved phytoliths were counted and classified to taxon; weathered phytoliths were classified as indeterminate. Except for one sample, all yielded statistically reliable phytolith counts averaging above 300 morphotypes.

## Microcharcoal

Each sample (2 ml) was mixed with 3% sodium hexametaphosphate (NaPO3)6 in a water bath (90 °C) for 30 min and mechanically agitated over several days. Lycopodium spore tables were added to calculate pollen and microcharcoal concentrations. Samples settled and decanted (<2 μm), followed by sieving (>125 μm). Separation of pollen and charcoal was done using gentle mechanical agitation over several days followed by heavy liquid separation using lithium heteropolytungstate at specific gravity 2.0. Acetolysis was achieved through a 9:1 mixture of $C_4H_6O_3$ to $H_2SO_4$ and 10% HCl. The samples were mounted on a

microscope slide with glycerol (cover: 22 × 40 mm = inspected area). System microscopy was done at 25x, 40x, 63x. A minimum of 350 pollen grains and 200 microcharcoal were counted.

## Paleosol geochemistry

**X-ray Fluorescence (XRF) and Loss on Ignition (LOI).** Bulk paleosols ($n = 22$) underwent major oxide concentration analysis using the XRF fusion bead method on a Zetium model spectrometer at Laboratório de Caracterização Tecnológica (LCT), Universidade de São Paulo (USP), Brazil. Samples (<200 mesh) were agate-milled to minimize metal contamination. About 1 g of finely powdered samples was fused with lithium tetraborate ($Li_2B_4O_7$) to form glass beads, eliminating particle size and matrix effects. Measurement accuracy, assessed with physical and analysis replicates and reference materials (GBW 07401, GBW 07404, Feldspar LCT-05, and Granite LCT-07), surpassed 5% for major oxides and ~7% for minor oxide $P_2O_5$. Precision, reflected by relative standard deviation (RSD), was 1% for major oxides and ~2.8% for minor oxide $P_2O_5$.

Loss on Ignition (LOI) estimation involved heating ~1 g of finely powdered samples at 1020 °C for 2 h using a muffle furnace, presenting LOI (wt%) = 100 × (ash weight1020 °C/sample weight).

**Inductively coupled plasma optical emission spectroscopy (ICP OES).** ICP OES analysis on selected pedogenic carbonate-free paleosols ($n = 8$) determined CaO* in silicates. Pedogenic carbonate removal involved treatment with 1 N HCl, followed by washing, drying, and multi-acid digestion (HF: HNO3: HClO4: HCl). A 300 mg powdered (<200 mesh) sample underwent digestion with 2.4 ml of HCl-HF-HNO3 mixture (2:2:1). Accuracy and precision for CaO* were 2.3% and 0.5% (RSD), respectively. The analysis was performed on an iCap 6300 Duo model Thermo Scientific spectrometer at LCT, USP, using standards GBW 07401 and GBW 07404 for quality control within the same digestion conditions.

## Reporting summary

Further information on research design is available in the Nature Portfolio Reporting Summary linked to this article.

## Data availability

All samples collected for paleomagnetic and environmental analyses are provided in the attached spreadsheets. Data on artifact types and raw material are also provided in the supplementary material. Three dimensional models of artifacts were captured under a NACOSTI permit from the Government of Kenya. The archeological and fossil fauna collection is housed in the Archeology Division of the National Museums of Kenya under the SASES description (FwJj 52). To access these materials contact the Head of the Archeology Section of the Earth Sciences Department of the National Museums of Kenya (Dr. Christine Ogola; cogola@gmail.com). All notes associated with the excavated materials have been provided to the museum. The analyzed measurements derived from three-dimensional models are available on request, due to museum restrictions. We provide the data associated with several other analyses. Analyses (e.g., Fig. 6) rely on data derived from previous publications see data provided in references[48,50,90–94]. We provide extensive bibliographic information regarding the location of data from previous publications on the technology of other Oldowan sites Supplementary Data 10.

## Code availability

All analyses used standard publicly available code and packages. Most analysis was conducted in the data analysis and visualization software R (version 4.03). When specific R packages were used for visualization or analysis they are described in detail. Details of specific software used for paleomagnetic analysis are described in detail in the Methods and Supplementary Information.

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

## Acknowledgements
We acknowledge the support and assistance provided by the Ileret community and the Archeology Department of the National Museums of Kenya during the course of this research. Funding for this study was generously provided by the U.S. National Science Foundation (NSF REU 1852441, NSF 1358178, NSF 1624398) the Paleontological Scientific Trust, and the Leakey Foundation. We are thankful to the National Museums of Kenya, the Kenyan National Commission for Science, Innovation, and Technology (NACOSTI), the Kenyan Department of Mining, and the Daasanach community for their collaboration. We are indebted to Purity Kiura, Robert Moru, and all participants of the Koobi Fora Field School for their research support. We also acknowledge Project PNRR C9 - I8 "Multiproxy reconstruction of Eurasian Megalakes, connectivity and isolation patterns during Neogene-Quaternary times", code 97/15.11.2022, Contract No. 760115/23.05.2023; the grant Veni.212.136 financed by the Dutch Research Council (NWO), grant 2018/20733-6 of Fundação de Amparo à Pesquisa do Estado de São Paulo (FAPESP), postdoctoral grant 2019/11364-0 of FAPESP and the American Museum of Natural History.

## Author contributions
D.R.B., D.V.P.R., N.G.B., A.K.B., R.B., F.F., A.S.H., R.N.K., A.M., E.K.N., D.B.P, J.S.R., M.J.S., K.T.U., A.V., J.G.W., J.W.K.H., W.A., and S.C. conceptualized the study. D.R.B., D.V.P.R., W.A., E.L.A., N.G.B., M.D.B., E.B., R.B., K.E., F.F., A.S.H., L.J., R.N.K., A.P.deM., P.R.D.M., A.M., J.M., E.K.N., D.B.P., J.S.R., D.C.R., M.J.S., P.S., K.T., K.T.U., A.V., J.G.W., J.W.K.H. and S.C. collected data. D.R.B., D.V.P.R., W.A., E.L.A., N.G.B., R.B., F.F., R.N.K., A.P.deM., P.R.D.M., J.M., E.K.N., D.B.P., J.S.R., D.C.R., M.J.S., P.S., K.T., K.T.U., A.V., J.G.W., J.W.K.H. and S.C. analyzed data. D.R.B., D.V.P.R., N.G.B., R.B., F.F., R.N.K., E.K.N., D.B.P., J.S.R., M.J.S., K.T., K.T.U., A.V., J.G.W., J.W.K.H. and S.C. wrote the paper. All authors discussed the results and commented on the manuscript.

## Funding

## Competing interests
The authors declare no competing interests.

## Additional information

David R. Braun [1,2] ✉, Dan V. Palcu Rolier [3,4,5] ✉, Eldert L. Advokaat [6], Will Archer[7,8,9], Niguss G. Baraki [2,10], Maryse D. Biernat[11], Ella Beaudoin[12], Anna K. Behrensmeyer[2,13], René Bobe [14,15,16], Katherine Elmes[17], Frances Forrest [18], Ashley S. Hammond [19,20,21], Luigi Jovane [5], Rahab N. Kinyanjui [22,23], Ana P. de Martini [5], Paul R. D. Mason [6], Amanda McGrosky [24,25], Joanne Munga[2], Emmanuel K. Ndiema[22], David B. Patterson[26], Jonathan S. Reeves [1,2,16], Diana C. Roman [27], Mark J. Sier [6,28,29], Priyeshu Srivastava [5,30], Kristen Tuosto[2,31], Kevin T. Uno [32,33], Amelia Villaseñor [34], Jonathan G. Wynn[35], John W. K. Harris[36] & Susana Carvalho [14,15,16,37]

[1]Technological Primates Research Group, Max Planck Institute for Evolutionary Anthropology, Leipzig, Germany. [2]Center for the Advanced Study of Human Paleobiology, George Washington University, Washington, DC, USA. [3]National Institute of Marine Geology and Geo-ecology, GeoEcoMar, Bucharest, Romania. [4]Fort Hoofddijk Paleomagnetic Lab, Department of Earth Sciences, Utrecht University, Utrecht, Netherlands. [5]Department of Geological Oceanography, Oceanographic Institute, University of São Paulo, São Paulo, Brazil. [6]Department of Earth Sciences, Utrecht University, Utrecht, Netherlands. [7]Max Planck Partner Group, Department of Archaeology and Anthropology, National Museum Bloemfontein, 9301 Bloemfontein, South Africa. [8]Florisbad Quaternary Research Station, National Museum Bloemfontein, 9301 Bloemfontein, South Africa. [9]Department of Geology, University of the Free State, Zastron Street, 9301 Bloemfontein, South Africa. [10]Department of Archeology and Tourism Studies, Addis Ababa University, Addis Ababa, Ethiopia. [11]Institute of Human Origins, School of Human Evolution and Social Change, Arizona State University, Tempe, Arizona, USA. [12]Department of Archaeology, University of Cambridge, Cambridge, UK. [13]Department of Paleobiology, Smithsonian National Museum of Natural History, Washington, DC, USA. [14]Department of Science, Gorongosa National Park, National Park, Sofala, Mozambique. [15]Primate Models for Behavioural Evolution Lab, University of Oxford, Oxford, UK. [16]ICArEHB, Universidade do Algarve, Faro, Portugal. [17]Department of Archaeology, University of Cape Town, Cape Town, South Africa. [18]Department of Sociology and Anthropology, Fairfield University, Fairfield, CT, USA. [19]Institut Català de Paleontologia Miquel Crusafont (ICP-CERCA), Universitat Autonoma de Barcelona, 08193 Cerdanyola del Valles, Barcelona, Spain. [20]Institució Catalana de Recerca i Estudis Avançats (ICREA), 08010 Barcelona, Spain. [21]Division of Anthropology, American Museum of Natural History, New York, NY 10024, USA. [22]Department of Earth Sciences, National Museums of Kenya, Nairobi, Kenya. [23]Department of Archaeology, Max Planck Institute for Geoanthropology, Jena, Germany. [24]Department of Evolutionary Anthropology, Duke University, Durham, NC, USA. [25]Department of Biology, Elon University, NC Elon, USA. [26]Department of Biology, University of North Georgia, Dahlonega, GA, USA. [27]Earth and Planets Laboratory, Carnegie Institution for Science, Washington, DC, USA. [28]CENIEH, Paseo de Atapuerca 3, Burgos, Spain. [29]Human Origins Research Unit, Faculty of Archaeology, Leiden University, Einsteinweg 2, 2333 CC Leiden, The Netherlands. [30]Indian Institute of Geomagnetism, Navi Mumbai, India. [31]Cultural Resources Management Department, San Manuel Band of Mission Indians, Highland, CA, USA. [32]Department of Human Evolutionary Biology, Harvard University, Cambridge, MA, USA. [33]Department of Earth and Planetary Sciences, Harvard University, MA 02138 Cambridge, MA, USA. [34]Department of Anthropology, The University of Arkansas, Fayetteville, AR, USA. [35]Directorate for Geosciences, National Science Foundation, Alexandria, VA, USA. [36]National Museums of Kenya, Archaeology Division, Nairobi, Kenya. [37]CIBIO/InBIO, Centro de Investigação em Biodiversidade e Recursos Genéticos, Universidade do Porto, Campus de Vairão, Vairão, Portugal. ✉e-mail: David_braun@gwu.edu; dan.palcu@gmail.com

