## [Transparent Peer Review file · Nature Communications]

Early Oldowan Technology Thrived During Pliocene Environmental Change in the Turkana Basin, Kenya

Corresponding Author: Dr David Braun

Part of this Peer Review File have been redacted as indicated to remove third-party material.

Version 0:

Reviewer comments:

Reviewer #1

(Remarks to the Author)

The authors present a newly discovered site, FwJj 52, Burgi Member, Koobi Fora Formation, Kenya. Three archaeological horizons are described from a critical time period in Oldowan technology, from 2.75-2.44 Ma. In this manuscript, the authors present tool making, geology, chronology, and paleoenvironmental descriptions and results. As the authors state, an issue with early Oldowan technology is that it is difficult to study changes in technological behavior, especially in response to environmental change. This is because the earliest Oldowan sites provide snapshots, but do not contain continuous deposits. The authors find and describe continuity in tool-making for 300,000 years. The argument presented in this manuscript is that there was climate-driven paleoenvironmental change at the end of the Pliocene, and that there is a relationship between this paleoenvironmental change and the development of stone tool technologies. In addition, the authors argue that there is paleoenvironmental change in the Turkana Basin from 2.75-2.44 Ma, yet the stone tool technology they describe remains stable during this 300,000 year time period. It is unclear whether the authors are arguing for or against technological innovation being linked to paleoenvironmental change, or why in some scenarios, for example at the end of the Pliocene, innovative technology is linked to paleoenvironmental change, whereas during the early Pleistocene there is a stasis in stone tool technology. The authors suggest that paleorivers are related to stone tool technology, and that once rivers replaced a lake on the landscape, stone tool technology persisted. This is explained well. However, it would be helpful to explain and expand on how local changes in vegetation, as evidenced from the well-described carbonates, plant wax biomarkers, phytoliths, and faunal remains, are related to the stasis in stone tool technology described by the authors.

In order to strengthen the paper, the authors should include section photographs with in situ artifacts. Wall sections with point provenience are shown in Figure 4, but photographs of artifacts in situ should also be included. If figure space is an issue, then perhaps section photographs with artifacts can be added to the supplementary material. It is essential in a paper describing some of the earliest excavated Oldowan technology. The authors refer to controversial stone tools from Lomekwi and cite replies that question the context of those artifacts. Section photographs of FwJj 52 with in situ artifacts should be included here. It would also be helpful to report the distance between the three archaeological sites.

The descriptions of stone tools and geology in the main text and supplementary material are thorough and presented well. This manuscript represents a large amount of fieldwork, expertise, and interdisciplinary collaboration. It is a welcome contribution. Some figure and table comments follow.

Figure and Table Comments:

Figure 1a should have a scale bar for the Turkana Basin.

Figure 6j is cropped at the base, Figure 6k the legend and label "k" are cropped and not visible. What do the boxes and crosses represent on Figure 6k? It is not indicated on the legend. In addition, for Figure 6k, the legend says this is showing change, but it looks like this figure is showing a snapshot of modeled climate, not change from time slice 1 to time slice 2. It is unclear if we are seeing modeled climate or which areas become more arid from the Pliocene to the Pleistocene. Please revise this figure as it is unclear from both the figure itself and the legend.

Supplementary Figure 6 a scale bar is needed on the microscope images, and the lowermost inset image is rectangular while the box is square on the lower color photograph. It looks like the second inset image is not indicated on the correct area on the color photograph, since we can see the edge of the bone on the inset image, but the box is not on the edge of the bone in the color photograph.

Minor comments: Supplementary Table 7 has spelling errors in the names of the fauna. Is "Aepyceotiini" referring to Aepycerotini? Some "sp." are italicized while others are not.

Reviewer #2

(Remarks to the Author)

Review:

The article supplies a 46m composite stratigraphic section that includes three new stratigraphy levels bearing Oldowan tools in Turkana basin (East Africa). One of these layers has also vertebrate fossils with evidence of butchery. These archaeological sites occurs on sediments that range between 2.75 and 2.44 Ma, a period that were previously thought to be absent in most of the northern portion of the Koobi Fora Fm.

The age model for this stratigraphic succession is based on the presence of the Tulu Bor tuff at the base of the sequence and four magnetic zones that place the sites in a very old period, up to 2.75 Ma for the oldest level. Two short reverse intervals are missing in this section probably because of erosion or included in sandstone beds not sampled in the paleomagnetic study. This study is exciting because corresponds to the equivalent time frame to the appearance of the genus Homo and the oldest Oldowan tools ever recovered. These tools represent the oldest tools in the Koobi Fora formation, which is very relevant for the area.

Different paleoclimatic proxies have been studied along 46 m of strata recording the period between 3.4 and 2.2Ma which is a period poorly understood in the basin because of large-scale erosional events. Within this record, three superposed strata dated at 2.75, 2.58, and 2.44 Ma show the continuity in Oldowan technology in the area for a period of 0,3 Ma. These proxies suggest a climatic shift at the time of the first appearance of Oldowan tools. This local paleoclimate record can be correlated with other paleoclimatic records in East Africa and Ocean cores, so the authors claim a causal connection between environmental shifts and the appearance of the Oldowan tool technology in the Turkana Basin. Although this is a speculative hypothesis, the presented data supports the contemporaneity between these two phenomena. Additional future data would validate or not this idea. I think the article supplies important archaeological, chronological, and paleoenvironmental information that deserves to be published in Nature Communications. However, I have a list of comments and suggestions that I think would be good to address before publication.

General Comments:

FwJ52 is an important archaeological site and will be cited by many colleagues in the future, so probably deserves a name easy to remember. So, if the name of this site has not been published before I suggest a simpler name, easier to remember and place on a map, if possible with a geographical reference, for example, the name of the closest town, Ileret52?, local topographic name.

Age model

It is not clear why you correlated NZ2 with the Feni 208 sub-chron (Reunion excursion) and not with Olduvai. Especially when considering the second scenario with Burgi Unconformity at the C7/L1 lithologic boundary, as previously suggested by Kidney23 and Baldes et al.26. This would solve two problems, a) sedimentation rate would be more reasonable for this interval, and B) you can correlate the upper lacustrine unit with Lorenyang lake that occurs at Olduvai and not with pre-Lorenyang lake. This does not necessarily change the age of your sites since an erosional surface exists at this boundary. Please explain better your reasons for this correlation and maybe include this possibility in your model that includes the Burgi Unconformity .

Methods: please specify from the 160 paleomagnetic samples you study, you should indicate the number of samples that correspond to a single strata (sampling station), and how many stratigraphic levels (sampling stations) you sample. Also, consider placing, supplementary material Figure 3 in the main text, since this is a very important figure.

In supplementary info: figures 18 and 19, cite the software you use to prepare the plots

In pmag supplementary tables: use only English language in the text

Minor comments:

Line 210: Mammoth and Kaena are periods with persistent reverse polarity during 84ka and 123ka, so you should use the term "subchron" or "subchronozone" not "excursion" for Mammoth and Kaena.

Line 293: is the shift towards arid conditions compatible with the lake phases between 2.2 and 2.4Ma shown in Fig 3 ? explain better

Line 427: since "Lorenyang lake" occurs in the Turkana lake system, I wonder why is not considered it just as Lake Turkana highstand? (Lorenyang highstand),

Line: 434 "During the late Pliocene, the EARS witnessed significant landscape and climate transformations such as lake regression, decreased woody cover, and aridification (Fig 6; e.g., 17,53,62–67) that significantly transformed vegetation communities, expanded grasslands, reduced woody cover "

No necessary to repeat "woody cover"

Line: 446 A pivotal event during this period was the closure of the Panama Seaway roughly between 2.8 to 2.5 million years ago, which had profound implications for Earth's climate and ecosystems 73

The reference indicated (73) suggests a late Miocene closure of the Panama isthmus (not Pliocene) based on U/Th zircons from Panama present in the North Andes, so rephrase please or add additional references.

Line 579 Using a 2.5 cm-diameter diamond-coated bit powered by a hand pump,

This sounds strange, usually battery or gasoline-powered drills are used to collect pmag samples, hand pump should refer to the water pump used to lubricate the bit.

Figure comments:

Figure 2: This is a nice figure, I suggest placing the stratigraphic sections in chronological order (A, B, C, D etc) and below the geological cross-section to read better the stratigraphy. I suggest placing the edges of the diagram vertically to improve the perspective.

Figure 3: the letters in the column are too small (ex TB).

Figure 4: I suggest indicating in the figure the % for tools of specific lithology, (ex. basalt 65%)

Figure 6: I suggest including the stratigraphic units at the base of the magnetic susceptibility plot, to see the relations of changes in different proxies with lithology.

In the plot "Water deficit vs age", the age numbers are cut. In the figure of "Plio-Pleistocene modeled climate", the legend can't be read well.

This figure has too much information, I suggest reducing it (50%) for example by leaving the data produced in this study (two columns with 3 plots each), and placing the other plots as Supplementary information.

Reviewer #3

(Remarks to the Author)

Braun and colleagues report on new archaeological sites in northern Kenya, in Area 40 of the Koobi Fora Formation, that span 2.75 to 2.44 Ma and are reported to the early Oldowan. By dealing with a time span that figures the earliest known development of a 'systematic production of sharp-edged tools', this paper adds significant insights to our understanding of the adaptation of early toolmakers to changing environments. This is obviously of great significance at both local and global scales. The Koobi Fora Formation, with its major unconformities, in particular the Burgi Unconformity which covers much of the 3 to 2 Ma time span, was the only formation of the Omo Group that lacked evidence for stone tool production prior to 2 Ma. Although we might of course have suspected that the early Oldowan groups that occupied the northern and western parts of the Turkana Basin (Shungura Fm and Nachukui Fm) might also have been present in its eastern part, it remained to be demonstrated that the Koobi Fora Fm had the potential to preserve late Pliocene / early Pleistocene artifact-bearing horizons. This is convincingly demonstrated for the first time in this article. This changes the paleogeography of the area, while confirming its status as a key area for exploring patterns of hominin behavioral adaptation during the earliest stages of the Oldowan. The succession of three archaeological horizons in site FwJi 52 of Area 40, located in the northern part of the Koobi Fora Formation, adds a temporal dimension with age estimates of 2.75, 2.58 and 2.44 Ma. This enables a discussion on patterns of behavioral continuity in a fluvial environment that has remained stable over this period. This article represents a huge amount of multidisciplinary analytical effort, for a new site complex that will definitely enrich our understanding of the early stages of the Oldowan technology. Our recommendations aim to better highlight the potential of these new sites for discussing patterns of hominin technological adaptation, based on the local environmental settings and associated resources at Koobi Fora and more broadly in the Turkana Basin, where discrete trajectories are documented. This would bring more originality to the article, which, despite presenting important new archaeological findings, tends to stick to long-debated ideas without providing groundbreaking evidence.

One of the main challenges posed by this article is the dating issue, as the marker layers that bracket the archaeological horizons define a particularly long time windows, ranging from 3.44 Ma to 2.2 Ma. Erosional phases and unconformities occurred within this time range, although it is not very clear how and where in the sequence they have been identified. The authors point out in p.5 that there is "a gap between 3.0 -2.5 Ma represented by the Burgi Unconformity" in Area 40, a statement which, unless explained, does not conform with what follows. Clarification about the Burgi Unconformity in this area is absolutely required to avoid confusion and to provide a solid basis for chronostratigraphic studies.

While we are not qualified to discuss the details of the paleomagnetic investigations, some precautions must be taken in the text; e.g. the age model, as robust as it appears, is based on age estimates, except for the Tulu Bor Tuff at the bottom of the sequence, and not on radiometric dates. For this reason, it should be more appropriate to use the term 'age estimates' rather than 'dates' or 'dated' (see for instance the Introduction). Similarly, the three chronological points inferred from the paleomagnetic data cannot be taken as local age markers equivalent to the Tulu Bor Tuff, but rather as derived or inferred benchmark ages.

Based on a combination of local sections and a synthetic stratigraphic log, the geological context is particularly well documented and illustrated for the entire FwJ52 area. At a more local scale, precision regarding the depositional environment of the archaeological sites is limited to their location in coarse sands within the gravel and paleosol package. Precisions regarding their position within the drainage system, and which drainage system (e.g. meandering, braided river?), could however be highly relevant for assessing the proximity of the sites to cobble-rich conglomerates that could have served as raw material supply areas. These data would enrich the arguments about raw material availability, as put forward in the discussion. They would also allow for more consistent insights into site taphonomy. We can infer from the sedimentological data that the archaeological remains are in secondary position. Demonstrating how the drainage system affected depositional and redepositional processes would therefore be highly relevant. For example, what explains the strong vertical dispersion of artifacts as seen in Fig.4, particularly in the older and younger sites? In this context, how can the artifacts be considered as deposited on a flat surface (SM Fig5)? What about assemblage integrity and artifact alteration, e.g.

degree of abrasion? What might explain limited size sorting and the absence or limited proportions of preferentially oriented artifacts in coarse sands (SM Fig5)?

The attribution of the assemblages to the early Oldowan seems well established, although pictures or drawings of artifacts illustrating a sample of flakes and cores are lacking to fully support this statement. The artifacts shown in Figure 4 lack descriptive captions and they are presented at too small a scale to be really informative. The relevance of data used to statistically discriminate Oldowan, Lomekwian and modern nonhuman primates must be justified, at least in Supplementary Material. More insight is also needed into the source data used in the PCA plots (see also below). Beyond their affiliation with the early Oldowan, what patterns of continuity can be observed between the three sites? Adding more detail on local chronological trends, or lack of thereof, will provide more insight into the notion of continuity beyond the long duration of the early Oldowan technology, which is already well established on a broader eastern African scale. Interpreting the data from KF Area 40 as indicative of continuity echoes what has previously been presented as Oldowan 'stasis' by Semaw and colleagues. Does the notion of continuity rather than stasis represent a paradigm shift, and on what basis, or is it simply a more neutral way of characterizing the enduring Oldowan technology? The overrepresentation of crypto-crystalline silica is of great interest. SM Fig. 11 shows that this pattern increases over time: does this raw material fit with distinctive morpho-functional attributes on flakes, e.g. in terms of cutting-edge extension? Given the role of raw material in this study, the inclusion of a detailed table showing the proportions of each raw material type across the sites would be beneficial (see also below).

The multi-proxy environmental studies indicate increased aridification between 2.7 and 2.2 Ma, a trend which is supported by a wealth of highly relevant data. This trend confirms what has already been extensively documented in other parts of the Turkana Basin. In the discussion, the authors jump from the local Koobi Fora scale to a global scale, which seems out of the scope of the article. It would make more sense, especially given the title of the article, to extend the scope to the Turkana Basin, where there is a wealth of paleontological, environmental and archaeological data available for this time window. Integrating the data from the neighboring Nachukui and Shungura formations would indeed show that, under the same environmental conditions, the earliest Oldowan occurrences only date to ca. 2.3 Ma in these contexts. Looking further south in the modern Lake Victoria margin, the Oldowan is documented since ca. 3.0-2.9 Ma at Nyayanga, well before the onset of climate aridification. In addition to the fact that the climate shift documented from ca. 2.8 Ma onwards appears to be more of a gradual process over hundreds of thousands years, its causal effect on the emergence of the Oldowan ultimately appears to be the less robust element of the discussion. This seems to be at least an oversimplification, while a more complex scenario, such as climatic aridification affecting resource availability differently in different parts of the Turkana Basin, could be discussed. In any case, more detail on the potential role of local raw material availability will strengthen the argument. This would once again require more detail on the drainage system and on site location within this system.

Below are a number of more targeted corrections and additions:

- The literature is not always appropriately cited. 1. The omission of Gona in the list of the 'earliest' early Oldowan sites, in the introduction, as well as in the discussion and conclusion, has to be corrected. Age estimates for Ledi Geraru and Gona largely overlap and there is no reason to consider one and not the other (Gona), which is furthermore the best documented Plio-Pleistocene site complex so far for the 3 to 2.5 Ma time range. 2. The references for the source data used in the main text (Fig. 5) and in the Supplementary Material (e.g. SM Figs 11 and 17) need to be cited, for instance in SM Table 8, even though these data are replicated from tables previously published by the authors. It is absolutely required for reproducibility purposes, especially since a quick check shows that the data from SM Table 8, that are used in the statistics from Fig.5, do not match, or only partly, with the published primary data, in particular for Gona, Nachukui and Shungura occurrences (we did not check the other site complexes). 3. The position of the references needs sometimes to be more appropriate, eg. 'These changes were instrumental in shaping the evolution and adaptations of both flora and fauna in eastern Africa as we show locally in this study 17,61,64,69,70': as obviously the references cited here do not proceed from this study, they have to be displaced after 'Africa'. 4. As a general rule, please cite first-hand data rather than second-hand ones.
- The authors' raw dataset that have been used for their quantitative analyses, including rock types, size and technological attributes, for the three assemblages from Area 40, should be compiled in an .xls file, as it is done for the paleomagnetic and paleoenvironmental datasets, again for reproducibility purposes.
- Figure 2 and SM Figures 2 & 3: the meaning of the abbreviations used in the stratigraphic columns, e.g. C1, C2, PO, P1, L1, etc. has to be detailed in the caption.
- Figure 4: the right panel is aesthetically interesting but confusing to read, with too many overlapping colors. The legend for product types and raw materials requires improvement. Its placement within the figure creates confusion regarding which elements correspond to each part.
- Figure 5: please enlarge the legend below the right panel
- Figure 6: the right side of the figure should be edited. The legend in Fig.6-K is cropped. The dates are also cropped in Fig.6-J.
- There are different notations for "STP_601", referred to as STP601 in some instances and as STP_601 in others. This should be homogenized.
- SM Figure 1: including the location of the sites alongside the geological sections would be beneficial.
- SM Figure 6: It would be helpful to specify the type of bone or clarify whether it is unidentified, rather than broadly referring to it as a 'portion of a size 3 suid'. The caption mentions four bones exhibiting evidence of butchery marks. For those not shown here, to which species do they belong, and what types are bones are they?
- SM Figure 19 is not mentioned in the main text.

(Remarks to the Author)

Version 1:

Reviewer comments:

Reviewer #1

(Remarks to the Author)

I have read through the revised manuscript and replies to reviewers. I think that the authors have done an impressive and detailed job addressing the concerns of the reviewers. Thank you for adding Supplementary Figure 6, showing in situ artifacts, and addressing all other comments I raised in a previous review. Figure 6 is a great improvement, although the x-axis labels are missing on Figures 6c, 6e, and 6f. Also, what does the vertical gray shading on Figures 6e, 6f, and 6g indicate? I think that the results of this study are noteworthy, and the claims are supported by the data presented. I recommend this manuscript for publication.

Reviewer #2

(Remarks to the Author)

Dear Editor,

The present version of the manuscript has covered all suggestions and questions presented in my previous review. The authors have modified the manuscript, making the paper stronger and easier to read. As I said before, the report of three new superposed archaeological levels at the Namorotukunan site dated between 2.7 and 2.4Ma is an important contribution, representing the earliest evidence of Oldowan technology within the Koobi Fora Formation and one of the oldest in the world. In addition, the article supplies important chronological and paleoenvironmental data for the studied period that place the finds in a paleoenvironmental context. For these reasons, I think this article deserves to be published in Nature Communications.

In my initial review, I was concerned about the age model, but the authors clarified my questions quite well in their answers and improved figures/text. I read carefully again the new version of the paper, and the cited references about the "Burgi Unconformity" (Kidney 2012, Baldes et al 2023 and Gathogo and Brown 2006). This has allowed me to have my personal opinion about the age and continuity of sedimentation in the studied area, which is a key question. By the end of my review, I suggest an important change in one paragraph that should be made to give consistency to the conclusions(see below).

Apart of that, I only found two minor typos in the Supplementary material easy to correct:

-Supplementary table 8: Perrisodactyla should be Perissodactyla, and this word appears separated in two parts as Cetartiodactyla.

-Supplementary table 9: check the references there are some typos, for example "de al Torre et al. 2003" should be "de la Torre et al. 2003"

Review of the Namorotukunan chrono-stratigraphy:

1-The presence of the Tulu Bor tuff dates the bottom of the section at 3.44Ma, this tuff is associated with normal magnetic polarity of NZ1, which after the Tulu Bor radiometric dates (3.44 Ma) corresponds to Gauss magnetozone. This normal zone is followed by a long reverse magnetic zone (RZ1) that should be Matuyama, so the Normal/Reverse polarity change identified in the lower part of the stratigraphy corresponds to the Gauss/Matuyama magnetic boundary (2.6Ma).

2-Three superposed archaeological sites appear below and above the Gauss/Matuyama boundary, allowing accuracy in the chronology of the three superposed archaeological sites. So, the main conclusion: "The site comprises three distinct archaeological horizons spanning approximately 300,000 years (2.75 - 2.44 Ma) within a persistent fluvial setting " is well supported by data.

3-The presence or absence of the Burgi Unconformity would affect only the age of strata in the uppermost part of the stratigraphy, above the archaeological sites, having a minimum effect on the chronology of the sites. But would not affect the continuity in the sedimentation around the sites all placed below the supposed unconformity.

4-The Burgi unconformity in the studied area:

The information about this unconformity in the study area (Namorotukunan, area 40) is not well documented. The unconformity is not mapped in Kidney 2012 map, where the contacts between Tulu Bor and upper Burgi Mbs are shown as conformable in his map (see below). In the studied area, the contact between Tulu Bor Mb and Burgi Mb is not erosional or angular, it is a plano-parallel contact. This plano-parallel contact is visible in the pictures presented by the authors of this study as well in other areas, >40 km south (Baldes et al 2023 of area 116). However, Gathogo and Brown mention a 20⁹

angular unconformity between these members in the Illeret area. Unfortunately, Gathogo and Brown 2006 do not show any field picture of this angular unconformity. From my field experience, faults and unconformities are very important structures that, if they exist, should be well documented because of their implication in the geological history of an area. According to the same authors, the sedimentary hiatus in area 40 represented at Burgi unconformity is close to 2Ma. "In Areas 40 and 41, the temporal gap is even larger, approaching 2 Ma, where the upper Burgi Member is in contact with the undifferentiated Moiti and Lokochot Members." (Gathogo and Brown 2006). We know after the present work that this is not correct because Tulu Bor tuff is in the section.

With the available information, I do not see any field evidence for the presence of an unconformity in the upper part of the studied section in Namorotukunan area. In contrast, the transition from alluvial to lacustrine units seems to be gradual according to figure 2, with an initial lacustrine incursion. The contact between Lake clays and underlying sediments is plano-parallel and does not show angular or erosional features. If someone claims this unconformity, here should name this boundary as a paraconformity and present evidence of a correlation with an angular/erosional unconformity elsewhere.

5-Age models

The authors propose two age models, one considers the presence of this unconformity and a sedimentary hiatus, and the other shows continuous sedimentation. If the unconformity is considered, the chronology of the two upper archaeological sites would be a bit older. The authors prefer the model without unconformity, because they have not seen the widely distributed KBS tuff (dated at 1,87 Ma and included in Olduvai magnetic subchron).

"In this paper, we use the first scenario, which provides more conservative (younger) age estimates. It is important to note that these scenarios do not affect the age estimation of the oldest archaeological levels presented in this paper"

According to Gathogo and Brown 2006 "In area 40 immediately north of Area 10, the KBS Tuff (K03-077; Table 2) is overlain by medium sandstones that fill channels oriented SSE–NNW" so this tuff seems to be associated in area 40 to fluvial deposits while in the studied area, the possible equivalent strata in the presence of unconformity, are lacustrine clays. The absence in these lacustrine deposits of the widely exposed KBS tuff (that occurs in Olduvai), supports the correlation of the upper normal event with Feni subchron.

With all the presented data, I agree with the preferred age model supporting a continuous sedimentation rate. However, a paragraph on line 135 introduces confusion, and it should be corrected.

Page 135:

"In Area 40, where Namorotukunan is located, sedimentary strata are estimated to range roughly between 4.3 to 1.6 Ma^{24,25}, with a gap between 3.0 -2.5 Ma represented by the Burgi Unconformity^{24,26,27}. Above this unconformity, a series of lake clays indicate a local change in tectono-sedimentary patterns and the Paleo-Lake Lorenyang transgression in the Turkana basin^{24,27}.

6-Comment:

This paragraph is included in the "Results/ Geological context" chapter and accepts the presence of the unconformity in area 40, covering in part the age of the new sites, which contradicts the selected age model. It is important to notice that from the references cited, the only one that directly includes the study of Namorotukunan area is Kidney 2012. But, although I did not have access to the full document (Phd), his map of the area shows a conformable contact between units, but not an unconformity (marked in geological maps with a dashed line). The other references do not include the Namorotukunan area. This sentence should be rewritten to avoid confusion. Part of it should be included in the introduction/geological setting, just to refer to the cited unconformity in the region. In the results chapter should be mentioned that no evidence of the Burgi Unconformity is found in the Namorotukunan section, which gives consistency with the selected age model.

I said before, the transition from alluvial to lacustrine units seems to be gradual in area 40, and can be related to wetter paleoclimate without involving tectonics and unconformity.

See Kidney 2012 map at
https://cdn.prod.websitefiles.com/5cc7bd30669af234434d1dca/5d266093b3094d49b7acf7f4_Kidney.pdf

Map published by Kidney 2012 showing a conformable contact between Tulu Bor (Tstb) and Upper Burgi (Tsub) Mbs around the Namorotukunan hill. In contrast, see unconformable contacts (dashed lines) for units Qs (Holocene) and T5In (early Pliocene).

Reviewer #3

(Remarks to the Author)

First of all, I would like to reiterate my appreciation for the substantial collective analytical effort presented in this article and to highlight the significant interest of the newly (re)investigated Area 40, located in the northern part of the Koobi Fora Formation. There is no doubt that the paleoenvironmental, stratigraphic, and archaeological data compiled in this study represent an important contribution to our understanding of the Early Pleistocene bio-environmental dynamics.

Although various minor aspects could still be discussed, my second review will focus exclusively on a single critical point: the chronological framework. This issue is pivotal, especially when addressing the earliest stages of human evolution. The challenge is particularly pronounced in this context due to the absence of radioisotopic dates that directly bracket the archaeological record combined with a discontinuous sedimentary sequence. Despite improvements in the manuscript and figures—especially Figures 2 and 3—and the inclusion of some details in both the main text and supplementary materials, key information is still missing.

In the Koobi Fora Formation, a major disconformity known as the Upper Burgi Unconformity has long been recognized by numerous researchers (e.g., Brown, Feibel, Kidney, Gathogo). According to the recent study by Baldes et al. (2023), this unconformity spans a time gap between approximately 2.70 and 2.08 Ma. This study is acknowledged briefly in the article (lines 141–142): “In Area 40, where Namorotukunan FwJj 52 is located, sedimentary strata are estimated to range roughly between 4.3 to 1.6 Ma, with a gap between 3.0 and 2.5 Ma represented by the Burgi Unconformity.” However, this important stratigraphic feature is not further discussed in detail, despite an apparent inconsistency with the age estimates proposed by Braun et al. (2.75, 2.58, and 2.44 Ma), which place the three archaeological sites squarely within the time gap identified by Baldes et al. for the Upper Burgi Unconformity (Area 116). A thorough discussion explaining this divergence in the proposed chronological framework is essential to substantiate the authors’ argument for “an enduring technological adaptation in the hominin lineage throughout the late Pliocene and earliest Pleistocene” (lines 128–129).

More specifically, the geochronological model presented by Braun et al. implies two potential scenarios:

- Scenario 1: the Upper Burgi Unconformity is absent in Area 40, which is the scenario used in the paper.
 - Scenario 2: the unconformity is present, marked by L1 bed, corresponding to the Lorenyang lake transgression as proposed by previous studies; the implication of this scenario, which is not discussed in the article, would be a time gap of ca. 0.3 Ma in Area 40, based on the 2.2 Ma age proposed by Braun et al. for the Lorenyang lake transgression (Lines 242–243). This differs significantly from Baldes et al. who place this boundary at 2.08 Ma in other parts of the Koobi Fora Fm, for a 0.62 Ma estimated time gap.
- o This second scenario opens the way for two options with regard to paleomagnetic correlations with lake transgression phases, implying a correlation of L1 with either the Pre-Lorenyang lake (option 1) or with Lorenyang lake (option 2), the first one being chosen because of the absence of the KBS tuff in the Area 40 sequence, which also differs from Baldes et al. who correlate L1 with the Lorenyang lake transgression.

This raises several important questions:

- If the Upper Burgi Unconformity is absent in Area 40 (Scenario 1), contra Baldes et al. (among others) who correlate it with the bottom of L1 bed, which lake transgression could L1 bed correspond to?
- What geomorphological or tectonic processes could account for continuous sedimentation in Area 40, in contrast to conclusions drawn by previous studies?
- Why is Scenario 1 (absence of unconformity) favored? Is the rationale that it “provides more conservative (younger) age estimates” (line 272) truly convincing or relevant in this context?
- If the unconformity is present (Scenario 2), how can you explain that the Upper Burgi Unconformity would be significantly shorter in Area 40 compared to other parts of the Koobi Fora Formation, such as Area 116?
- In a discontinuous sedimentary sequence, due to lake transgression phases, is it relevant to use the absence of chronological markers (e.g., KBS tuff) for establishing chrono-stratigraphic correlations?
- In the absence of an uppermost tephra layer, why not consider a third scenario—i.e., a longer-duration unconformity placing the new sites higher in the stratigraphic sequence, perhaps within the KBS Member?

Ultimately, regardless of the scenario selected, the chronostratigraphic model proposed in this article diverges significantly from previously established frameworks. This alternative model must be explicitly compared and contrasted with earlier work. With the evidence currently available, non-specialists in Koobi Fora chrono-stratigraphy are not in a position to confidently evaluate the robustness of the proposed age model. Given the significance of the archaeological sites and associated paleoenvironmental data, strengthening the chronological framework should be a top priority before drawing behavioral or paleoenvironmental inferences.

Reviewer #4

(Remarks to the Author)

Detailed response to reviews comments for “Early Oldowan Technology Thrived During Pliocene Environmental Change in the Turkana Basin, Kenya”

David R. Braun^{1,*}, Dan V. Palcu^{2,3,4,*}, Eldert L. L. Advokaat⁵, Will Archer^{6,7}, Niguss G. Baraki^{8,9}, Maryse Biernat¹⁰, Ella Beaudoin¹¹, Anna K. Behrensmeyer¹², René Bobe^{13,14,15}, Katherine Elmes¹⁶, Frances Forrest¹⁷, Ashley S. Hammond¹⁸, Luigi Jovane⁴, Rahab N. Kinyanjui^{19,20}, Ana P. de Martini⁴, Paul Mason⁵, Amanda McGrosky²¹, Joanne Munga⁸, Emmanuel K. Ndiema¹⁹, David B. Patterson²², Jonathan S. Reeves¹, Diana C. Roman²³, Mark J. Sier^{5,24}, Priyeshu Srivastava^{4,25}, Kristen Tuosto^{8,26}, Kevin Uno²⁷, Amelia Villaseñor²⁸, Jonathan G. Wynn²⁹, John W. K. Harris³⁰, Susana Carvalho^{13,14,15,31}

Reviewer #1 (Remarks to the Author):

Comment 1.1 (Reviewer 1) It is unclear whether the authors are arguing for or against technological innovation being linked to paleoenvironmental change, or why in some scenarios, for example at the end of the Pliocene, innovative technology is linked to paleoenvironmental change, whereas during the early Pleistocene there is a stasis in stone tool technology. The authors suggest that paleorivers are related to stone tool technology, and that once rivers replaced a lake on the landscape, stone tool technology persisted. This is explained well. However, it would be helpful to explain and expand on how local changes in vegetation, as evidenced from the well-described carbonates, plant wax biomarkers, phytoliths, and faunal remains, are related to the stasis in stone tool technology described by the authors.

Response to reviewer comment: *We appreciate your observation regarding the relationship between technological innovation and paleoenvironments, particularly the contrast between the innovation observed at the end of the Pliocene and the stasis during the early Pleistocene. To clarify, we argue that the emergence and persistence of the Oldowan tool technology were closely tied to specific local and regional environmental changes. Our revised text emphasizes the nuanced interaction between geological factors (availability of large cobbles and paleoriver systems) and paleoecological conditions (changes in the availability of resources). By presenting evidence from the Namorotukunan area (FwJj 52), we demonstrate how relatively dry climatic episodes facilitated artifact production and burial. Regarding the connection between vegetation shifts and the stasis in stone tool technology, we expanded our discussion to highlight the role of vegetation and its influence on hominin foraging strategies and tool use. Using data from carbonates, plant wax biomarkers, phytoliths, and faunal remains, we propose that the relative drying trend in vegetation observed locally during the early Pleistocene may have contributed to the appearance of stone tool technology in this area at this time due to paleoecological and paleogeographical shifts. These changes in the ancient ecosystems likely provided consistent resources, which, coupled with the availability of raw materials and depositional conditions, supported a continuity in Oldowan tool use .[Main text line 1268-1277]*

Comment 1.2 (Reviewer 1) In order to strengthen the paper, the authors should include section photographs with in-situ artifacts. Wall sections with point provenience are shown in Figure 4, but photographs of artifacts in situ should also be included. If figure space is an issue, then perhaps section photographs with artifacts can be added to the supplementary material. It is essential in a paper describing some of the earliest excavated Oldowan technology.

Response to reviewer comment: *We have now provided a supplementary figure (Supplementary Figure 6) that provides images of artifacts in situ showing their position at the*

time of their discovery and also provides images of these artifacts in detail.[Supplementary Materials]

Comment 1.3 (Reviewer 1) The authors refer to controversial stone tools from Lomekwi and cite replies that question the context of those artifacts. Section photographs of FwJj 52 with in situ artifacts should be included here. It would also be helpful to report the distance between the three archaeological sites.

Response to reviewer comment: *We now provide details of the size of the area around the archaeological sites. We also provide exact distances between sites. These are also provided in Figure 2 (see green lines between sections with distances). In addition, the distances between sites are visible in Supplementary Figure 1, which provides an overview map of the region. We also provide images of artifacts in situ showing the relationship between artifacts and the stratigraphy indicating that the majority of artifacts are over a meter below the modern surface [Supplementary Materials and Figure 2]*

Comment 1.4 (Reviewer 1) Figure 1a should have a scale bar for the Turkana Basin.

Response to reviewer comment: *We have modified Figure 1a to include a scale bar for the Turkana Basin and we have updated this figure to reflect the shifts in the names of the excavations as requested by Reviewer 2.[Line 232-243]*

Comment 1.5 (Reviewer 1) Figure 6j is cropped at the base, Figure 6k the legend and label “k” are cropped and not visible. What do the boxes and crosses represent on Figure 6k? It is not indicated on the legend. In addition, for Figure 6k, the legend says this is showing change, but it looks like this figure is showing a snapshot of modeled climate, not change from time slice 1 to time slice 2. It is unclear if we are seeing modeled climate or which areas become more arid from the Pliocene to the Pleistocene. Please revise this figure as it is unclear from both the figure itself and the legend.

Response to reviewer comment: *This figure has been completely re-organized based on multiple reviewer suggestions. We have now included details that are related to specific parts of the Turkana Basin and we have modified Figure 6 to make it clearer and more focused on local and regional shifts.[Line 731-740]*

Comment 1.6 (Reviewer 1) Supplementary Figure 6 a scale bar is needed on the microscope images, and the lowermost inset image is rectangular while the box is square on the lower color photograph. It looks like the second inset image is not indicated on the correct area on the color photograph, since we can see the edge of the bone on the inset image, but the box is not on the edge of the bone in the color photograph.

Response to reviewer comment: *This figure [now Supplementary Figure 8]has been revised so that the shape and orientation of the boxes that are placed on the large-scale images of the bone are the same as the microscope images. These microscope images are made with a spinning disc Nanofocus Microscope. As a result, the “edge” of the microscope image represents the edge of the 3D model that the microscope creates and not the actual edge of the*

bone. This is also the reason that scale bars cannot be accurately placed because the image reflects an oblique image of the surface of the bone which provides the ability to see internal striations on the marks (a hallmark of cut marks).

Comment 1.7 (Reviewer 1) Minor comments: Supplementary Table 7 has spelling errors in the names of the fauna. Is “Aepyceotiini” referring to Aepycerotini? Some “sp.” are italicized while others are not.

Response to reviewer comment: *We have modified this table to correct these mistakes.[Supplementary Materials]*

Reviewer #2 (Remarks to the Author):

Comment 2.8 (Reviewer 2) FwJj52 is an important archaeological site and will be cited by many colleagues in the future, so probably deserves a name easy to remember. So, if the name of this site has not been published before I suggest a simpler name, easier to remember and place on a map, if possible with a geographical reference, for example, the name of the closest town, Ileret52?, local topographic name.

Response to reviewer comment: *Thank you for your thoughtful comment regarding the naming of FwJj52 and its importance as a future reference point for colleagues. We appreciate the suggestion to adopt a simpler, geographically meaningful name for the site.*

In response, we have revised the site name to "Namorotukunan," which is a local topographic name closely associated with the area. Additionally, we have redefined the names of the individual archaeological excavations within the site to ensure clarity and consistency. These are now named as follows:

Site Name: Namorotukunan (we retain the FwJj52 moniker as an official name because it is the name that the National Museums of Kenya uses to classify the site.)

Archaeological Excavations:

- Namorotukunan1 (NMT1 – STP601 - the oldest excavation)*
- Namorotukunan2 (NMT2 – KT_016-05 - the second oldest excavation)*
- Namorotukunan3 (NMT3 – KT_016-04 - the youngest excavation)*

These changes have been incorporated into the manuscript to align with the suggestion for a simpler and more memorable naming convention, while maintaining ties to the geographical and cultural context of the site.

We hope this addresses your concern and enhances the clarity and accessibility of our work for future reference.[These changes can be found throughout the manuscript]

Comment 2.8 (Reviewer 2) : It is not clear why you correlated NZ2 with the Feni 208 sub-chron (Reunion excursion) and not with Olduvai.

Epecially when considering the second scenario with Burgi Unconformity at the C7/L1 lithologic boundary, as previously suggested by Kidney23 and Baldes et al.26. This would solve two problems, a) sedimentation rate would be more reasonable for this interval, and B) you can correlate the upper lacustrine unit with Lorenyang lake that occurs at Olduvai and not with pre-Lorenyang lake.

This does not necessarily change the age of your sites since an erosional surface exists at this boundary. Please explain better your reasons for this correlation and maybe include this possibility in your model that includes the Burgi Unconformity .

Response to reviewer comment: Thank you for your insightful comment regarding the correlation of NZ2 with the Feni 208 sub-chron (Reunion excursion) and your suggestion to consider its correlation with the Olduvai sub-chron. We appreciate your perspective and recognize the importance of addressing alternative interpretations to enhance the robustness of our analysis.

In response to your feedback, we have incorporated the following changes into the manuscript: We have added a detailed discussion of the alternative scenario that correlates NZ2 with the Olduvai sub-chron. Specifically, we considered the implications of placing the Burgi Unconformity at the C7/L1 lithologic boundary. We suggest that absence of the KBS Tuff in the Area 40 sequence (a widespread marker that is also visible directly to the south across the Gele fault - See Kidney 2012) makes it likely that this upper normal magnetozone is not the Olduvai sub-chron. However, we mention in the text that it is not possible to rule this option out. [Line334- 341]

To reflect this alternative interpretation, we have updated the relevant figure to include the Burgi Unconformity scenario [Figure 3]. The revised figure now clearly depicts both the original correlation with the Feni 208 sub-chron and the alternative correlation with the Olduvai sub-chron. Annotations and explanations have been added to ensure clarity and to facilitate comparison between the two models.

We believe these additions enhance the clarity and scientific rigor of the manuscript by presenting a more comprehensive analysis of the possible correlations. We hope this addresses your concerns effectively and demonstrates our commitment to thoroughly evaluating alternative hypotheses.

Comment 2.10 (Reviewer 2): Methods: please specify from the 160 paleomagnetic samples you study, you should indicate the number of samples that correspond to a single strata (sampling station), and how many stratigraphic levels (sampling stations) you sample.

Response to reviewer comment: Thank you for your valuable feedback and for highlighting the need for additional clarity in the methods section regarding the paleomagnetic samples.

*In response, we have updated the text to provide a more detailed explanation of the sampling strategy (in the Supplementary Materials pgs. 1-3). Specifically, we have clarified that the 160 paleomagnetic samples correspond to distinct stratigraphic levels (sampling stations). Each sample represents a unique stratigraphic level, ensuring that the dataset provides a comprehensive representation of the stratigraphic sequence studied. **We have added more information to clarify that the 160 samples correspond to 147 distinct stratigraphic levels with 13 doubles (two destroyed during measurements).** Additionally, we have indicated the number of stratigraphic levels sampled to further enhance transparency in the methods.*

Comment 2.11 (Reviewer 2): Also, consider placing, supplementary material Figure 3 in the main text, since this is a very important figure.

Response to reviewer comment: We have now modified Figure 2 so that it incorporates the details of Supplementary Figure 3. We have maintained Supplementary Figure 3 as a reference. We expect readers to use Supplementary Figure 3 to focus exclusively on the paleomagnetic results. However, most of the details provided in Supplementary Figure 3 can now be found in Figure 2.

Comment 2.12 (Reviewer 2): In supplementary info: figures 18 and 19, **cite the software you use to prepare the plots. In pmag supplementary tables: use only English language in the text**

Response to reviewer comment: Thank you for your comments and suggestions regarding the supplementary information and tables. We have updated the supplementary information to cite the software (FORCinel and paleomagnetism.org) used to prepare Supplementary Figures 18 and 19 (now Supplementary Figure 20 and 21). Specifically, we clarified that the plots were created using an online Java-based application, paleomagnetism.org, and included additional details about the software and its features to ensure transparency. Additionally, the text in the paleomagnetic supplementary tables has been reviewed and revised to ensure that all entries are in English. These corrections improve the clarity and accessibility of the supplementary materials, and we appreciate your feedback in helping us enhance the quality of the manuscript.

Comment 2.13 (Reviewer 2): Line 210: Mammoth and Kaena are periods with persistent reverse polarity during 84ka and 123ka, so you should use the **term “subchron” or “subchronozone” not “excursion” for Mammoth and Kaena.**

Response to reviewer comment: Thank you for this suggestion. We have now modified the text and Figures to use the term subchron for references to the Mammoth and Kaena subchrons.[line 329-330]

Comment 2.14 (Reviewer 2): Line 293: is the shift **towards arid conditions compatible with the lake phases between 2.2 and 2.4Ma shown in Fig 3 ? explain better**

Response to reviewer comment: We cannot find the specific text reference indicated in this comment within the manuscript. We have now modified the text to describe the paleoecology of the region in a way that describes various events as the result of local and regional events. The changes in the local ecosystem associated with the onset of lake phases at the top of the sequence may be associated with major lake phases described previously (e.g. Nutz et al., Boes et al.) however as these shifts occur after the archaeological horizons they do not influence our discussion of the interaction between ecosystems and the appearance of the Oldowan record. [Line 1282-1306]

Comment 2.15 (Reviewer 2): Line 427: since “Lorenyang lake” occurs in the Turkana lake system, I wonder why is not considered it just as **Lake Turkana highstand? (Lorenyang highstand),**

Response to reviewer comment: Thank you for your thoughtful comment regarding the terminology used to describe the Lorenyang Lake in the Turkana lake system. We acknowledge that referring to it as the “Lake Turkana highstand” (or “Lorenyang highstand”) is a valid alternative and part of a broader, ongoing discussion.

In response, we have opted to retain the term “Paleo-Lake Lorenyang transgression in the Turkana basin” as it is clearer and we found Lorenyang is commonly employed in the literature (Boes et al. 2018). However, we have added clarifications in the text to acknowledge this ongoing debate and to provide context for our choice. We hope this additional clarification addresses your concerns and provides a balanced perspective.[Line 537]

Comment 2.16 (Reviewer 2): Line: 434 “During the late Pliocene, the EARS witnessed significant landscape and climate transformations such as lake regression, decreased woody cover, and aridification (Fig 6; e.g., 17,53,62–67) that significantly transformed vegetation communities, expanded grasslands, reduced woody cover “

Response to reviewer comment: *Thank you for this astute correction, we have removed the term woody cover in the second instance of this term in this sentence. [Line 1230]*

Comment 2.17 (Reviewer 2): Line: 446 A pivotal event during this period was the closure of the Panama Seaway roughly between 2.8 to 2.5 million years ago, which had profound implications for Earth's climate and ecosystems 73

The reference indicated (73) suggests a late Miocene closure of the Panama isthmus (not Pliocene) based on U/Th zircons from Panama present in the North Andes, so rephrase please or add additional references.

Response to reviewer comment: *Thank you for bringing this to our attention. We have revised the text to clarify the timing and implications of the closure of the Panama Seaway. Initially, we referenced several sources to provide a broader context for this event, but in the updated text, we have refined our focus to the specific closure event around 2.75 million years ago. The references now align exclusively with this timeframe to ensure accuracy and consistency with the evidence. We appreciate your careful review and believe this revision resolves the issue effectively.[Line 1290]*

Comment 2.18 (Reviewer 2): Line 579 Using a 2.5 cm-diameter diamond-coated bit powered by a hand pump. This sounds strange, usually battery or gasoline-powered drills are used to collect pmag samples, **hand pump should refer to the water pump used to lubricate the bit.**

Response to reviewer comment: *Thank you for pointing this out. We appreciate the opportunity to clarify our methodology. In this case, the hand pump referred to was used to air-cool the samples during drilling, as the unconsolidated nature of the samples prevented the use of liquid for cooling the drill bit. This approach was chosen to minimize the risk of altering or damaging the samples due to heat generated during the drilling process. We have revised the text to make this distinction clear and to avoid any potential confusion.[Line 1471-1474]*

Comment 2.19 (Reviewer 2): Figure 2: This is a nice figure, I suggest placing the stratigraphic sections in chronological order (A, B, C, D etc) and below the geological cross-section to read better the stratigraphy. I suggest placing the edges of the diagram vertically to improve the perspective.

Response to reviewer comment: *Thank you for your suggestions regarding Figure 2. We have made the recommended changes. The stratigraphic sections are now arranged in chronological order (A, B, C, D, etc.). They are placed below the geological cross-section to improve readability. We have also adjusted the edges of the diagram to be vertical for better perspective. Additionally, we have completely revised the figure to enhance clarity and visual presentation. We appreciate your feedback and believe the updated figure addresses your concerns effectively.[Figure 2]*

Comment 2.20 (Reviewer 2): Figure 3: the letters in the column are too small (ex TB).

Response to reviewer comment: *As described above this figure was completely redesigned which included increasing text size for readability.[Figure 3]*

Comment 2.21 (Reviewer 2): Figure 4: I suggest indicating in the figure the % for tools of specific lithology, (ex. basalt 65%)

Response to reviewer comment: Thank you for your suggestion regarding Figure 4. We appreciate your attention to detail. The figure originally included a graphical representation of the percentages of each raw material. We have now added text that explicitly describes the percentages for specific lithologies, such as basalt (e.g., XX%). This is now incorporated into the visual representation. However, we have ensured that these percentages are written out and easy to interpret for the reader.[Figure 4]

Comment 2.22 (Reviewer 2): Figure 6: I suggest including the stratigraphic units at the base of the magnetic susceptibility plot, to see the relations of changes in different proxies with lithology.

In the plot “Water deficit vs age”, the age numbers are cut. In the figure of “Plio-Pleistocene modeled climate”, the legend can’t be read well.

Response to reviewer comment: Thank you for this suggestion. We have added a plot of lithology to Supplementary Figure 17 to address this point. This plot highlights the stratigraphic units and provides clarity on the relationship between magnetic susceptibility changes and lithological variations. We have also included additional explanations in the figure caption and the main text to link the observed magnetic susceptibility changes with specific lithological changes. Figure 6 has also been completely revised to make it easier to read.

Comment 2.23 (Reviewer 2): This figure has too much information, I suggest reducing it (50%) for example by leaving the data produced in this study (two columns with 3 plots each), and placing the other plots as Supplementary information.

Response to reviewer comment: Thank you for your feedback regarding the complexity of this figure. We understand your concerns and have addressed them by reorganizing and correcting the figure. Specifically, we now include data exclusively focused on the Turkana Basin to reduce the amount of information presented at once. We increased the font size to facilitate readability. These changes should make the figure clearer and more accessible while retaining the critical information. Additional plots not directly produced in this study have been moved to a Supplementary Figure to streamline the main figure.[Figure 6]

Comment 3.24 (Reviewer 3): This article represents a huge amount of multidisciplinary analytical effort, for a new site complex that will definitely enrich our understanding of the early stages of the Oldowan technology. Our recommendations aim to better highlight the potential of these new sites for discussing patterns of hominin technological adaptation, based on the local environmental settings and associated resources at Koobi Fora and more broadly in the Turkana Basin, where discrete trajectories are documented.

This would bring more originality to the article, which, despite presenting important new archaeological findings, tends to stick to long-debated ideas without providing groundbreaking evidence.

Response to reviewer comment: We have now modified aspects of the manuscript (in response to multiple reviewer comments) to focus more exclusively on the local and regional changes in the Namorotukunan region and the impact on the appearance of stone artifacts in this part of the record. We highlight contextual variables within this region and its influence on tool use. [These changes occur in many places within the manuscript but specifically in Lines:128-155; 481-485; 1275-1285]

Comment 3.25 (Reviewer 3): One of the main challenges posed by this article is the dating issue, as the marker layers that bracket the archaeological horizons define a particularly long time windows, ranging from 3.44 Ma to 2.2 Ma.

Erosional phases and unconformities occurred within this time range, although it is not very clear how and where in the sequence they have been identified. The authors point out in p.5 that there is “a gap between

3.0 -2.5 Ma represented by the Burgi Unconformity” in Area 40, a statement which, unless explained, does not conform with what follows. Clarification about the Burgi Unconformity in this area is absolutely required to avoid confusion and to provide a solid basis for chronostratigraphic studies.

Response to reviewer comment: Thank you for your insightful comment regarding the Burgi Unconformity and its implications for the dating of the archaeological horizons. We agree that identifying and clarifying the Burgi Unconformity is critical for ensuring a solid basis for chronostratigraphic studies.

In response, we confirm that we do indeed identify the Burgi Unconformity in the sequence, and this is explicitly illustrated in Figure 3 of the manuscript (as indicated previously by Baldes et al. 2024) in the Namorutukanan region. We also recognize the importance of this unconformity in interpreting the stratigraphy and its potential to represent large time gaps. However, as noted by Reviewer 1, even with an alternative explanation that the Burgi Unconformity represents a significant period of time (e.g., 3.0–2.5 Ma), this does not affect the ages of the archaeological sites. All sites are situated stratigraphically below the unconformity, meaning that their placement in the sequence is unaffected by the presence or interpretation of this erosional surface. The confirmation of the Gauss-Matuyama boundary below the Burgi Unconformity as well as the well dated Tulu Bor Tuff at the base of the section provides upper and lower boundaries to create a chronostratigraphic model of these sediments. We have revised the text to clarify the location and implications of the Burgi Unconformity in Area 40, ensuring that its identification and significance are clearly explained. We hope these adjustments address your concerns and improve the clarity and robustness of the manuscript. [Line 225-228;334-341]

Comment 3.26 (Reviewer 3): While we are not qualified to discuss the details of the paleomagnetic investigations, some precautions must be taken in the text, e.g. the age model, as robust as it appears, is based on age estimates, except for the Tulu Bor Tuff at the bottom of the sequence, and not on radiometric dates.

For this reason, it should be more appropriate to use the term ‘age estimates’ rather than ‘dates’ or ‘dated’ (see for instance the Introduction). Similarly, the three chronological points inferred from the paleomagnetic data cannot be taken as local age markers equivalent to the Tulu Bor Tuff, but rather as derived or inferred benchmark ages.[3]

Response to reviewer comment: We appreciate the reviewer’s thoughtful comments regarding the terminology used to describe chronological markers and age estimates. Specifically, we acknowledge that paleomagnetic shifts provide robust chronological markers and understand the importance of referring to them as relative age markers rather than absolute dates. We have made the changes in the text as well as figures. We have replaced references to ‘dates’ or ‘dated’ with ‘age estimates’ where appropriate, and we emphasize that the age model is indeed based on inferred age estimates rather than radiometric dates, except for the Tulu Bor Tuff at the bottom of the sequence [Line 93 and 353].

Comment 3.27 (Reviewer 3): At a more local scale, precision regarding the depositional environment of the archaeological sites is limited to their location in coarse sands within the gravel and paleosol package. Precisions regarding their position within the drainage system, and which drainage system (e.g. meandering, braided river?), could however be highly relevant for assessing the proximity of the sites to cobble-rich conglomerates that could have served as raw material supply areas.

These data would enrich the arguments about raw material availability, as put forward in the discussion. They would also allow for more consistent insights into site taphonomy. We can infer from the sedimentological data that the archaeological remains are in secondary position.

Response to reviewer comment: We appreciate the reviewer's suggestion to provide greater precision regarding the depositional environment of the archaeological sites and their relationship to the surrounding drainage systems. In response, we have expanded our description to include detailed information about the types of river systems associated with the sites. Specifically, we now identify whether the archaeological horizons are associated with braided river systems, as well as the geomorphic context of the sites within these systems.

Additionally, we have clarified that these river systems are directly adjacent to the archaeological horizons. This revised discussion highlights the proximity of the sites to cobble-rich conglomerates that could have served as raw material sources, providing further context to support our arguments about raw material availability in the discussion [Line 479-486].

We agree that these data also have important implications for understanding site taphonomy. As indicated by our site formation analysis, the sites have been impacted by higher energy deposition. However, this was not extensive enough to remove the smaller fraction of the flaked materials (Supplementary Figure 4) or orient the artifacts that were deposited (Supplementary Figure 5).

Comment 3.28 (Reviewer 3): Demonstrating how the drainage system affected depositional and redepositional processes would therefore be highly relevant. For example, what explains the strong vertical dispersion of artifacts as seen in Fig.4, particularly in the older and younger sites?

In this context, how can the artifacts be considered as deposited on a flat surface (SM Fig5)? What about assemblage integrity and artifact alteration, e.g. degree of abrasion? What might explain limited size sorting and the absence or limited proportions of preferentially oriented artifacts in coarse sands (SM Fig5)?

Response to reviewer comment: Thank you for your detailed comment regarding the role of the drainage system and its impact on depositional and post-depositional processes. We appreciate the opportunity to clarify these points and address your concerns.

The vertical dispersion of artifacts, as seen in Figure 4, is representative of the extent of coarse-grained deposits at each locality. Specifically, the vertical dispersion at all three sites reflects the distribution and thickness of these coarse-grained depositional units. Importantly, at all of the localities, artifacts are present in the stratigraphic sequence when rocks of a suitable size (to make stone artifacts) are also deposited in these sequences. This suggests a strong association between artifact deposition and the availability of suitable sedimentary contexts.

Artifacts are considered to have been deposited on a relatively flat surface, as indicated by the orientation analysis (Supplementary Material Figure 5). While artifact orientations align with the dominant river direction, their flat-lying positions suggest deposition occurred on multiple aggrading surfaces rather than erosional (or subsequent redeposited) surfaces. These aggrading surfaces facilitated the rapid burial of artifacts by coarse-grained deposits, preserving them in situ and maintaining assemblage integrity. Some size sorting is indicated by the absence of the smallest fraction seen in the NMT3 assemblage (Supplementary Figure 4). This interpretation is further supported by the absence of clear erosional features associated with these depositional surfaces.

Regarding assemblage integrity and artifact alteration, we observe limited evidence of abrasion, which supports the interpretation that artifacts were rapidly buried after deposition. Similarly, the limited size sorting and absence (or minimal proportions) of preferentially oriented artifacts within coarse sands reflect the dynamic but localized depositional environment of these aggrading surfaces. Together, these findings indicate that the artifacts were not significantly reworked after deposition, preserving the integrity of the assemblages. [Lines 362-426]

We have incorporated these clarifications into the manuscript to address these points and better illustrate the depositional and post-depositional processes at play.

Comment 3.29 (Reviewer 3): The attribution of the assemblages to the early Oldowan seems well established, although pictures or drawings of artifacts illustrating a sample of flakes and cores are lacking to fully support this statement.

The artifacts shown in Figure 4 lack descriptive captions and they are presented at too small a scale to be really informative.

Response to reviewer comment: *Thank you for the suggestion of increasing the number of artifacts that can be displayed. We have now added a subsequence Supplementary Figure 7. This provides more images of the artifacts at a larger scale.[Supplementary Figure 7]*

Comment 3.30 (Reviewer 3): The relevance of data used to statistically discriminate Oldowan, Lomekwian and modern nonhuman primates must be justified, at least in Supplementary Material.

Response to reviewer comment: *Thank you for your comment regarding the justification of the data used to statistically discriminate between Oldowan, Lomekwian, and modern non-human primate artifacts. We appreciate the opportunity to clarify this important aspect of the study.*

The data used to discriminate between Lomekwian and Oldowan artifacts are based on variables that have been extensively described and utilized in previous studies (Braun et al. 2019; Plummer et al. 2023). These variables are grounded in the original publications that first characterized the Lomekwian (Harmand et al. 2015) and Oldowan (Leakey 1973) technologies, reflecting key morphological and technological patterns distinguishing these assemblages. Similarly, the differences between non-human primate artifacts and Oldowan tools follow the patterns described in the original publications (Proffitt et al. 2023) that documented and analyzed non-human primate artifacts.

Our study builds on this foundational work by providing a statistical framework to quantitatively distinguish these assemblages. The measurements we used are those explicitly described in the original studies (Harmand et al. 2015; Braun et al. 2019; Plummer et al. 2023) and were chosen specifically because they have been demonstrated to distinguish between these groups. We have added clarifications in the Supplementary Material to ensure that the basis for these measurements is clear and explicitly tied to the existing literature. We have also provided a new table (Supplementary Table 9) which provides detailed references for each measurement.

Given the robust statistical differences demonstrated between the groups using these measurements, we are confident that the data are both relevant and appropriate for the purpose of this analysis. If there are additional concerns or specific aspects that reviewers feel need further justification, we would be happy to address them in more detail.

We further thank the reviewer for noting possible discrepancies in the Nachukui assemblages. We note that we followed the protocol described by Braun et al. 2019. That protocol describes an imputation process so that portions of the dataset that were not available in the literature were replaced via imputation based on the surrounding data. Given that some of the data from the Lokalelei 2C assemblage was imputed we have chosen to remove this assemblage from the analysis. The resulting visualization shows similar patterns to the analysis previously presented [Figure 5].

Comment 3.31 (Reviewer 3): Beyond their affiliation with the early Oldowan, what patterns of continuity can be observed between the three sites? Adding more detail on local chronological trends, or lack thereof, will provide more insight into the **notion of continuity beyond the long duration of the early**

Oldowan technology, which is already well established on a broader eastern African scale. Interpreting the data from KF Area 40 as indicative of continuity echoes what has previously been presented as Oldowan ‘stasis’ by Semaw and colleagues.

Response to reviewer comment: We appreciate the reviewer’s insightful comments on the patterns of continuity and local chronological trends across the sites. In our revised manuscript, we emphasize that our analyses of technological attributes reveal no statistically significant differences in artifact morphology between the different horizons at Namorotukunan. This finding highlights that the materials from the three excavations cannot be distinguished based on the technological attributes analyzed in this study.

In response to the comment regarding local chronological trends. We do not explicitly refer to the concept of stasis within the Oldowan but rather describe the technological patterns through time. A broader study of change throughout the Oldowan would require data from across numerous Oldowan sites, while controlling for raw material availability and quality. This data is not currently available. While the early Oldowan’s long duration is well established on an eastern African scale, we interpret the data from Namorotukunan as providing evidence of local continuity [Figure 5; Supplementary Figure 9, 11, 18]. Here we only emphasize that these assemblages are most similar to the Early Oldowan assemblages. We have integrated this perspective into our discussion to provide a more nuanced interpretation of local technological trends within our dataset [Line 1300-1319; 1421-1432.

Comment 3.32 (Reviewer 3): Does the notion of continuity rather than stasis represent a paradigm shift, and on what basis, or is it simply a more neutral way of characterizing the enduring Oldowan technology? The overrepresentation of crypto-crystalline silica is of great interest. SM Fig. 11 shows that this pattern increases over time: does this raw material fit with distinctive morpho-functional attributes on flakes, e.g. in terms of cutting-edge extension?

Response to reviewer comment: We appreciate the reviewer’s thought-provoking question regarding whether the notion of continuity rather than stasis represents a paradigm shift or simply a more neutral characterization of the enduring Oldowan technology. In our manuscript, we do not aim to propose a paradigm shift from previous interpretations of technological competence in the Oldowan. Rather, we describe the patterns as they appear in the archaeological record. Our analyses indicate no technological differences between the horizons, supporting a consistent pattern of technological behavior over time.

Regarding the overrepresentation of crypto-crystalline silica and its increase over time, we agree that this is a compelling observation. We have examined whether this raw material correlates with distinctive technological attributes on flakes through the sequence. Our analysis, however, finds no evidence of such a correlation within the dataset. This suggests that the preference for crypto-crystalline silica was not tied to specific technological characteristics of the tools. All of the chalcedony flakes in the 3 layers appear to be similar in shape and size. A more detailed analysis of the raw material properties and their relationship to specific morpho-functional attributes is beyond the scope of this study.

Comment 3.33 (Reviewer 3): Given the role of raw material in this study, the inclusion of a detailed table showing the proportions of each raw material type across the sites would be beneficial (see also below).

Response to reviewer comment: Thank you for your helpful comment regarding the inclusion of detailed raw material data. In response, we have included this information in the revised version of Figure 5, which now provides a clear visual representation of the proportions of each raw material type across the sites. Additionally, we have attached spreadsheets containing the raw material attributions for all artifacts analyzed in this study. These additions ensure transparency and allow readers to fully engage with the raw material data presented in our analysis.

Comment 3.34 (Reviewer 3): The multi-proxy environmental studies indicate increased aridification between 2.7 and 2.2 Ma, a trend which is supported by a wealth of highly relevant data.

This trend confirms what has already been extensively documented in other parts of the Turkana Basin. In the discussion, the authors jump from the local Koobi Fora scale to a global scale, which seems out of the scope of the article. It would make more sense, especially given the title of the article, to extend the scope to the Turkana Basin, where there is a wealth of paleontological, environmental and archaeological data available for this time window.

Response to reviewer comment: *Thank you for your detailed feedback and constructive suggestions. We appreciate your emphasis on the importance of contextualizing our study within the broader framework of the Turkana Basin.*

In response, we have modified Figure 6 to include information from the entire Turkana Basin, encompassing data from the Nachukui Formation and other key components of the Omo Group. This revised figure provides a more comprehensive representation of the multi-proxy environmental studies across the Basin and underscores the broader trend of increased aridification between 2.7 and 2.2 Ma, supported by extensive paleontological, environmental, and archaeological data.

To address your concern about the scope of the discussion, we have revised this section to focus more explicitly on the Turkana Basin rather than making broad global-scale inferences. This adjustment aligns with the title of the article and highlights the wealth of data from the Turkana Basin that substantiates the observed aridification trend. By doing so, we aim to strengthen the connection between our findings and the broader regional context while maintaining the article's central focus [Line 736 – 1319].

Comment 3.35 (Reviewer 3): Integrating the data from the neighboring Nachukui and Shungura formations would indeed show that, under the same environmental conditions, the earliest Oldowan occurrences only date to ca. 2.3 Ma in these contexts.

Response to reviewer comment: *See our response regarding Figure 6 and our focus on Figure 6 and the associated details of the Omo Group.*

Comment 3.36 (Reviewer 3): Looking further south in the modern Lake Victoria margin, the Oldowan is documented since ca. 3.0-2.9 Ma at Nyayanga, well before the onset of climate aridification. In addition to the fact that the climate shift documented from ca. 2.8 Ma onwards appears to be more of a gradual process over hundreds of thousands years, its causal effect on the emergence of the Oldowan ultimately appears to be the less robust element of the discussion.

Response to reviewer comment: *Thank you for your insightful comment regarding the timing of the Oldowan's emergence in relation to global climate shifts. We have addressed this in our revised text by incorporating the following points:*

First, the Nyayanga material from the modern Lake Victoria margin is dated to between 2.6 and 3.0 Ma. Plummer et al 2023 acknowledge that the large error bars of the radiometric dates only allow an estimation of “early within the temporal range of the C2An.1n Subchron” (Plummer et al. 2023: 563) which currently dates to 2.581 - 3.032 Ma. This time range overlaps with the timing of the 2.75 Ma shift in global climates. It is not possible to definitively confirm whether the Nyayanga artifacts were created at the time of major climatic shifts or not. Thus the influence of this global climate transition on the appearance of stone artifacts in eastern Africa cannot be investigated using the Nyayanga materials.

*Second, our revised text emphasizes that the appearance of artifacts at Namorotukunan is associated with a **local shift in ecosystems** rather than being directly tied to global trends. While we discuss the potential influence of global climate changes, we stress that these shifts are more likely related to regional dynamics within the basin. Specifically, we highlight that local environmental changes, combined with the availability of suitable raw materials for tool production, played a key role in the emergence of stone artifacts in this area. [Line 1310-1319;1421-1432]*

By focusing on these local factors and providing explanations for how regional environmental changes may reflect broader climatic processes, we aim to present a balanced and nuanced perspective. We appreciate your feedback, which has helped us refine our discussion and clarify these important points.

Comment 3.37 (Reviewer 3): This seems to be at least an oversimplification, while a more complex scenario, such as climatic aridification affecting resource availability differently in different parts of the Turkana Basin, could be discussed. In any case, more detail on the potential role of local raw material availability will strengthen the argument. This would once again require more detail on the drainage system and on site location within this system.

Response to reviewer comment: *We appreciate the reviewer's insightful comment regarding the complexity of environmental factors and the role of local raw material availability. In our revised manuscript, we have provided additional detail about the drainage systems and their relationship to raw material availability at the sites. Specifically, we describe the geomorphic context of the sites within these systems and discuss how proximity to raw material sources may have influenced technological behaviors.*

We also emphasize the importance of local paleogeographic shifts. While our data highlight patterns specific to this portion of the Turkana Basin, we recognize that these dynamics could vary significantly across different regions of the basin or other parts of Africa. This variability underscores the need to consider both local and regional factors when interpreting patterns in the archaeological record.

By integrating these points, we aim to present a more nuanced understanding of how local raw material availability and environmental changes may have shaped early Oldowan technological behaviors.

Comment 3.38 (Reviewer 3): The omission of Gona in the list of the 'earliest' early Oldowan sites, in the introduction, as well as in the discussion and conclusion, has to be corrected. Age estimates for Ledi Geraru and Gona largely overlap and there is no reason to consider one and not the other (Gona), which is furthermore the best documented Plio-Pleistocene site complex so far for the 3 to 2.5 Ma time range.

Response to reviewer comment: *We have now revised the manuscript to include references to Gona in all mentions of the earliest Oldowan assemblages. This ensures that the discussion acknowledges the importance of Gona as a key site in understanding the origins and early development of Oldowan technology. We appreciate your feedback, which has allowed us to improve the completeness and accuracy of our references [Line 464-486; Line 764-1278].*

Comment 3.39 (Reviewer 3): 2. The references for the source data used in the main text (Fig. 5) and in the Supplementary Material (e.g. SM Figs 11 and 17) need to be cited, for instance in SM Table 8, even though these data are replicated from tables previously published by the authors. It is absolutely required for reproducibility purposes, especially since a quick check shows that the data from SM Table 8, that are used in the statistics from Fig.5, do not match, or only partly, with the published primary data, in particular for Gona, Nachukui and Shungura occurrences (we did not check the other site complexes).

Response to reviewer comment: We appreciate the reviewer's detailed feedback regarding the references and reproducibility of the data used in the manuscript, particularly in relation to Table 8, Fig. 5, and the Supplementary Material. To address these concerns, we have added citations for all source data used in the analysis. These are outlined in Supplementary Table 9. This ensures full transparency and reproducibility by clearly identifying the studies from which the variables were derived.

The values in Table 8 have been cross-checked against those reported in the original publications, and we believe they accurately reflect the primary data. However, if there are any remaining discrepancies, we are happy to revise the table to incorporate updated or corrected data as new publications become available.

Regarding Lokalalei 2C, we have excluded this site from the canonical analysis because imputing the large number of missing variables would have compromised the robustness of the analysis.

These revisions aim to ensure that the data are accurately represented, reproducible, and aligned with the original sources. We thank the reviewer for highlighting these points and helping us improve the clarity and reliability of the manuscript. [Supplementary Table 9, 10]

Comment 3.40 (Reviewer 3): 3. The position of the references needs sometimes to be more appropriate, e.g. 'These changes were instrumental in shaping the evolution and adaptations of both flora and fauna in eastern Africa as we show locally in this study 17,61,64,69,70': as obviously the references cited here do not proceed from this study, they have to be displaced after 'Africa'. 4. As a general rule, please cite first-hand data rather than second-hand ones.

Response to reviewer comment: We have modified the references to conform with standard procedures. We appreciate the reviewer identifying these concerns [Supplementary Table 9,10].

Comment 3.41 (Reviewer 3): - The authors' raw dataset that have been used for their quantitative analyses, including rock types, size and technological attributes, for the three assemblages from Area 40, should be compiled in an .xls file, as it is done for the paleomagnetic and paleoenvironmental datasets, again for reproducibility purposes.

Response to reviewer comment: Thank you for your comment regarding the inclusion of raw datasets for the artifact assemblages. We have provided an .csv file containing artifact numbers along with basic technological information, including rock types, and tool type, for the three assemblages from Namorotukunan. This level of detail surpasses the data that has been previously made available for other Oldowan sites. We believe this dataset enhances transparency and reproducibility and is a much higher standard of data sharing than has been provided for other stone artifact assemblages (Supplementary Table 6).

Comment 3.42 (Reviewer 3): - Figure 2 and SM Figures 2 & 3: the meaning of the abbreviations used in the stratigraphic columns, e.g. C1, C2, PO, P1, L1, etc. has to be detailed in the caption.

Response to reviewer comment: Thank you for pointing out the need for clarity regarding the abbreviations used in Figure 2 and Supplementary Figures 2 and 3. We have revised the captions to include these details. We also make reference to Supplementary Table 2, where all the abbreviations (e.g., C1, C2, PO, P1, L1, etc.) are defined and their descriptions are detailed. This ensures that readers have access to the necessary information to interpret the stratigraphic columns accurately. We appreciate the reviewer highlighting this and have taken steps to make the figures and their captions more transparent and user-friendly.

Comment 3.43 (Reviewer 3): - Figure 4: the right panel is aesthetically interesting but confusing to read, with too many overlapping colors. The legend for product types and raw materials requires improvement. Its placement within the figure creates confusion regarding which elements correspond to each part.

Response to reviewer comment: Thank you for your thoughtful feedback on Figure 4. The right panel is a **Sankey plot**, a type of visualization routinely used in social science data to represent complex relationships between multiple groups. We have now added details to ensure clarity, including providing the raw data used to generate the plot. Readers can calculate the actual values from the raw data included in the supplementary materials.

Additionally, we have improved the legend for product types and raw materials by reformatting its placement to minimize confusion [Figure 4].

Comment 3.44 (Reviewer 3): - Figure 5: please **enlarge the legend below the right panel**

Response to reviewer comment: Thank you for your comment regarding Figure 5. We have made the requested changes by enlarging the legend below the right panel to improve its readability [Figure 5].

Comment 3.45 (Reviewer 3): - Figure 6: the **right side of the figure should be edited**. The legend in Fig.6-K is cropped. The dates are also cropped in Fig6-J.

Response to reviewer comment: This change has been noted by other reviewers and this figure has been completely revised in the resubmitted version [Figure 6].

Comment 3.46 (Reviewer 3): - There are different notations for "STP_601", referred to as STP601 in some instances and as STP_601 in others. This should be homogenized.

Response to reviewer comment: The multiple different uses of the terms for excavation is now negated because of the revised names for all excavations as noted in our previous response to reviewer 1's comment about the names of the site and the excavations [Multiple changes throughout the text].

Comment 3.47 (Reviewer 3): - SM Figure 1: including the location of the sites alongside the geological sections would be beneficial.

Response to reviewer comment: Supplementary Figure 1 has been revised to incorporate the new names used in the entire manuscript as per the revision suggested by Reviewer 1.

Comment 3.48 (Reviewer 3): - SM Figure 6: It would be helpful to specify the type of bone or clarify whether it is unidentified, rather than broadly referring to it as a 'portion of a size 3 suid'. The caption mentions four bones exhibiting evidence of butchery marks. For those not shown here, to which species do they belong, and what types are bones are they?

Response to reviewer comment: Thank you for your comment regarding the identification of bone types in the figure. The type of bone is already mentioned in the caption, but we recognize the need for additional clarity. In response, we have modified the caption to include descriptions of the types of other bones with modifications that were recovered during the excavations. This added detail ensures a more comprehensive and transparent presentation of the data [Supplementary Data].

Comment 3.49 (Reviewer 3): - SM Figure 19 is not mentioned in the main text.

Response to reviewer comment: Supplementary Figure 19 (now supplementary Figure 21) has now been incorporated into the text.

Response to the Editor and Reviewers

We thank the editor and reviewers for their detailed feedback, which helped improve the clarity and robustness of our chronostratigraphic framework. We have carefully revised the manuscript to better align our interpretation of the geological and magnetic polarity data with current regional stratigraphic models, particularly as they relate to the KBS Tuff and the Upper Burgi Unconformity (UBU).

Reviewer 1 Comment: Reviewer 1 provided encouragement and support that our revised manuscript provides a clearer and more robust understanding of the context and location of artifacts at the NMT sites.

Reviewer 2 Comment: The second reviewer made some minor suggestions about typographical errors and incomplete references. We have made the adjustments in the revised text.

Reviewer 2 also mentions that after reviewing the paleomagnetic and stratigraphic information, they believe that the chrono-stratigraphy we present is well supported by the data. Reviewer 2 also reviews the current description of the Burgi Unconformity in the Koobi Fora Fm. They indicate that based on current photographic evidence in all current publications there is no convincing evidence for an actual angular unconformity. The reviewer confirms that our age estimate is a conservative estimate and that other photographic evidence does not provide adequate information necessary to support a major unconformity. The reviewer suggests that current evidence does not support the presence of a significant unconformity. They suggest that we indicate this sedimentary sequence as conformable given the gradual transition over to lacustrine sediments above the G/M boundary.

We have opted to provide an age model that includes multiple possible scenarios (as indicated in Figure 3). Importantly, the age of NMT1, and NMT 2 are not affected by this because they sit adjacent to or below the G/M boundary. Therefore, as the potential Burgi Unconformity only influences the age of the youngest archaeological site we opted to provide the conservative age estimate. As indicated in our response to Reviewer 3, we believe we have provided the most conservative age model. Furthermore providing multiple possible scenarios allows for the opportunity for multiple interpretations of the sedimentary sequence.

Reviewer 3 Comment: Stratigraphic correlation and absence of KBS Tuff

We acknowledge the reviewer's concern about the correlation of the Namorotukunan sequence to the broader Koobi Fora stratigraphy, especially regarding the presence of the KBS Tuff and the UBU. In response, we have provided a more detailed discussion of the geological, lithostratigraphic, and paleomagnetic context in the revised manuscript.

The KBS Tuff is an important and regionally extensive stratigraphic marker, and its absence at Namorotukunan plays a key role in our interpretation. We now emphasize that the KBS Tuff has been described ~600 meters west of Namorotukunan, across the North Gele Fault (previously mapped and samples by Kidney 2012), where it overlies a thick succession of lacustrine clays. This lithologic package differs markedly from the Namorotukunan stratigraphy, which ends in a thinner lake clay sequence, lacking both the KBS Tuff and the associated lithologies. This strongly suggests that the Namorotukunan sequence does not extend into the KBS Member.

Reviewer 3 Comment: Interpretation of the Burgi Unconformity

We have expanded our discussion of the Burgi Unconformity (UBU) in light of its expression—or lack thereof—in Area 40. Although we acknowledge that many of the volcanic ashes in the Turkana Basin are variably expressed (lateral variability in the thickness of tuffs is noted throughout the Turkana Basin). However, the fact that the KBS tuff is expressed as a highly tuffaceous unit (>2m thick) directly south of the Tulu Bor and Burgi Member deposits in Area 40. As Kidney states in his 2012 description of the area: “The KBS member must have been deposited in the study area, however after faulting occurred, subsequent erosion removed the KBS member strata east of the North Gele fault.”(page 108). The UBU has been described as marking a significant depositional hiatus in other parts of the Koobi Fora Formation. However, in Area 40, the transition from fluvial to lacustrine deposits is conformable, and we find no sedimentological evidence for an erosional surface at this contact. We therefore present two scenarios in the revised manuscript:

- Scenario A: Continuous sedimentation across the transition.
- Scenario B: A brief and localized hiatus, potentially represented by a subtle disconformity not apparent in the field.

Both scenarios are compatible with the archaeological and paleoenvironmental evidence and do not alter the placement or interpretation of the artifact-bearing horizons. We have also clarified that if the UBU is present, it must be restricted to a short interval within RZ1, implying a much briefer gap than that documented further south in Koobi Fora.

Our data indicate that sedimentation in Namorotukunan resumed during an interval when other parts of the basin were still within the unconformity stage. This indicates that, if an unconformity occurred here, it ended earlier and was shorter than at other sites. In the B scenario, we have used the published 2.5 Ma age for the onset of the unconformity; however, if the upper bound differs, it is equally reasonable to question whether the lower bound is the same. Without direct constraints on its true duration, any age model incorporating this unconformity would rely on arbitrary assumptions. For this reason, scenario A, the linear interpolation model remains the most stable and defensible option, and its use does not affect the paper’s key findings.

Reviewer 3 Comment: Correlation of magnetic polarity zones

We have revised the magnetostratigraphic interpretation to make clear that R1 corresponds to the Matuyama Chron, and that NZ2 is more parsimoniously correlated with the Feni excursion (~2.12–2.14 Ma) rather than the Olduvai subchron (~1.95–1.78 Ma). This interpretation is based on several lines of evidence:

- The absence of the KBS Tuff in the lacustrine sediments associated with NZ2.
- The stratigraphic position of NZ2 above the Gauss–Matuyama boundary.
- The lithological mismatch between the Namorotukunan sequence and the lacustrine clay package associated with the KBS Member immediately to the west of the Gele fault.

While we acknowledge the possibility that NZ2 could correspond to the Olduvai subchron, we find this less likely given the local sedimentological context and the lack of tephrostratigraphic markers supporting such a correlation.

Summary of Key Revisions in the Manuscript

- We now clearly situate the KBS Tuff ~600 m west of Namorotukunan and contrast the stratigraphy across the North Gele Fault.
- We provide a two-scenario model for the expression (or absence) of the UBU in Area 40.
- We offer a reasoned argument favoring correlation of NZ2 with the Feni excursion, consistent with the paleomagnetic, stratigraphic, and tephrostratigraphic data.
- We explicitly state that both scenarios result in consistent placement of the archaeological sites between ~2.75 and 2.44 Ma, during the late Pliocene and early Pleistocene.

We trust these clarifications address the reviewer's concerns and help support the internal consistency of our geochronological model.

Smaller editorial revisions. The reviewers indicate that the x-axis was missing in figure 6 e, f, g. However, the x-axis on this figure is time. We have aligned all sections of this figure to align with time. We have provided the age on two of the figures, but we feel as though repeating the same axis 5 times would be repetitive.

The reviewers note that the name of the Order Perissodactyla has been spelled incorrectly in Supplementary Table 8. We have edited this, and we have also edited the misspelling of names in Supplementary Table 10.

Reviewer #2 cross-comments to Reviewer #3

Reviewer #3

First of all, I would like to reiterate my appreciation for the substantial collective analytical effort presented in this article and to highlight the significant interest of the newly (re)investigated Area 40, located in the northern part of the Koobi Fora Formation. There is no doubt that the paleoenvironmental, stratigraphic, and archaeological data compiled in this study represent an important contribution to our understanding of the Early Pleistocene bio-environmental dynamics.

-I agree with this

Although various minor aspects could still be discussed, my second review will focus exclusively on a single critical point: the chronological framework. This issue is pivotal, especially when addressing the earliest stages of human evolution. The challenge is particularly pronounced in this context due to the absence of radioisotopic dates that directly bracket the archaeological record combined with a discontinuous sedimentary sequence. Despite improvements in the manuscript and figures—especially Figures 2 and 3—and the inclusion of some details in both the main text and supplementary materials, key information is still missing.

-After the review of all cited references, I have not found a specific paper about the chronology of area 40, so I think the new data improve the chronology of this area in Koobi Fora.

In the Koobi Fora Formation, a major disconformity known as the Upper Burgi Unconformity has long been recognized by numerous researchers (e.g., Brown, Feibel, Kidney, Gathogo). According to the recent study by Baldes et al. (2023), this unconformity spans a time gap between approximately 2.70 and 2.08 Ma.

This study is acknowledged briefly in the article (lines 141–142): “In Area 40, where Namorotukunan FwJj 52 is located, sedimentary strata are estimated to range roughly between 4.3 to 1.6 Ma, with a gap between 3.0 and 2.5 Ma represented by the Burgi Unconformity.” However, this important stratigraphic feature is not further discussed in detail, despite an apparent inconsistency with the age estimates proposed by Braun et al. (2.75, 2.58, and 2.44 Ma), which place the three archaeological sites squarely within the time gap identified by Baldes et al. for the Upper Burgi Unconformity (Area 116). A thorough discussion explaining this divergence in the proposed chronological framework is essential to substantiate the authors’ argument for “an enduring technological adaptation in the hominin lineage throughout the late Pliocene and earliest Pleistocene” (lines 128–129).

-Baldes et al 2023 do not study the area 40 (Namorotukunan area), their data comes mainly from area 116, which is >40km away. The field pictures presented in the article show conformable strata, no unconformable strata. In case of clear evidence of sedimentary hiatus, this boundary should be named a paraconformity to be more concise.

-Different from the submitted paper that supply the position of Tulu Bor tuff and paleomagnetic boundaries, Baldes et al 2023 use only the lithological contact with units of different color to map the cited unconformity, but no chronological data are supplied below and above this plano-parallel contact to justify a sedimentary hiatus (see published fig.6).

“Light Tulu Bor Member deposits (blue) contrast with dark brown Upper Burgi clays (green). Purple line marks the contact used to map the UBU (fig 6)” .

[FIGURE REDACTED]

I have seen many vertical changes of facies (lake transgressions) like this in similar Pliocene-Pleistocene deposits from Ethiopia and Kenya triggered by climatic processes, not by tectonics, that do not imply any relevant sedimentary hiatus.

Fig. 6 from Baldes et al 2023 in area 116 40km away from Namorotukuna

More specifically, the geochronological model presented by Braun et al. implies two potential scenarios:

- Scenario 1: the Upper Burgi Unconformity is absent in Area 40, which is the scenario used in the paper.
- Scenario 2: the unconformity is present, marked by L1 bed, corresponding to the Lorenyang lake transgression as proposed by previous studies; the implication of this scenario, which is not discussed in the article, would be a time gap of ca. 0.3 Ma in Area 40, based on the 2.2 Ma age proposed by Braun et al. for the Lorenyang lake transgression (Lines 242-243). This differs significantly from Baldes et al. who place this boundary at 2.08 Ma in other parts of the Koobi Fora Fm, for a 0.62 Ma estimated time gap.

I don't see that Braun et al propose a 2.2 Ma age for the Lorenyang lake transgression in Lines 242-243. In scenario B the authors place the lacustrine claystones during Olduvai times, which is a time younger than 2Ma, see fig.3.

240 We present two alternative age models based on the different correlations described above. These age
241 models reflect two possible scenarios: the first scenario (Fig. 3a) assumes continuous sedimentation
242 between gravel levels C7 and lake clays L1, while the second scenario (Fig. 3b) incorporates the Burgi
243 Unconformity at the C7/L1 lithologic boundary, as previously suggested by Kidney²⁴ and Baldes et al.²⁷.
244 This second scenario results in slightly older age estimates for sites Namorotukunan-3 (NMT3) and
245 Namorotukunan-2 (NMT2). In this paper, we use the first scenario, which provides more conservative
246 (younger) age estimates. It is important to note that these scenarios do not affect the age estimation of the
247 oldest archaeological levels presented in this paper.

o This second scenario opens the way for two options with regard to paleomagnetic correlations with lake transgression phases, implying a correlation of L1 with either the Pre-Lorenyang lake (option 1) or with Lorenyang lake (option 2), the first one being chosen because of the absence of the KBS tuff in the Area 40 sequence, which also differs from Baldes et al. who correlate L1 with the Lorenyang lake transgression.

This raises several important questions:

- If the Upper Burgi Unconformity is absent in Area 40 (Scenario 1), contra Baldes et al. (among others) who correlate it with the bottom of L1 bed, which lake transgression could L1 bed correspond to?

-Baldes et al 2023 do not study the area 40 (Namorotukunan area), their data comes mainly from area 116 >40km away.

- What geomorphological or tectonic processes could account for continuous sedimentation in Area 40, in contrast to conclusions drawn by previous studies?

After this new work, area 40 has more magnetostratigraphic data, and the previous interpretation should be reviewed, I have not worked in the area, but the first option to explore to explain this lake transgression would be wetter paleoclimate.

- Why is Scenario 1 (absence of unconformity) favored? Is the rationale that it “provides more conservative (younger) age estimates” (line 272) truly convincing or relevant in this context?

I think there are different reasons to favor scenario 1, the most important is that there is no field evidence of an unconformity in the studied section. The second is the absence of KBS tuff, which occurs in the area in fluvial (not lacustrine) deposits.

- If the unconformity is present (Scenario 2), how can you explain that the Upper Burgi Unconformity would be significantly shorter in Area 40 compared to other parts of the Koobi Fora Formation, such as Area 116?

This should be studied; many explanations are possible

- In a discontinuous sedimentary sequence, due to lake transgression phases, is it relevant to use the absence of chronological markers (e.g., KBS tuff) for establishing chrono-stratigraphic correlations?

The absence of a chronostatigraphic marker in lacustrine claystones is informative data, especially when this marker is widely distributed and is present in area 40 in fluvial sediments (not in lacustrine), (Gathogo and Brown 2006).

- In the absence of an uppermost tephra layer, why not consider a third scenario—i.e., a longer-duration unconformity placing the new sites higher in the stratigraphic sequence, perhaps within the KBS Member?

The new sites are below the unconformity, near the Gauss/Matuyama boundary, so the duration of this possible hiatus located above the sites, it is not relevant for their chronology.

Ultimately, regardless of the scenario selected, the chronostratigraphic model proposed in this article diverges significantly from previously established frameworks.

In my opinion, the chronostatigraphy of the Namorotukunan area was not studied in detail, and the newly available data solves the age, especially for the lower part of the stratigraphy

This alternative model must be explicitly compared and contrasted with earlier work. With the evidence currently available, non-specialists in Koobi Fora chrono-stratigraphy are not in a position to confidently evaluate the robustness of the proposed age model. Given the significance of the archaeological sites and associated paleoenvironmental data, strengthening the chronological framework should be a top priority before drawing behavioral or paleoenvironmental inferences.

I'm a non-specialist in Koobi Fora chrono-stratigraphy, but after reviewing the available published information presented in the manuscript and references, I have a quite clear idea about the chrono-stratigraphy of Namorotukunan. Of course, future work can supply new data and refine this new chronostratigraphic frame.